# Scaling Laws Meet Model Architecture: Toward Inference-Efficient LLMs

**Song Bian**[*]    **Tao Yu**[†]      **Shivaram Venkataraman**    **Youngsuk Park**
UW-Madison    Amazon Web Services    UW-Madison      Amazon Web Services
{songbian,shivaram}@cs.wisc.edu, {taou,pyoungsu}@amazon.com

## ABSTRACT

Scaling the number of parameters and the size of training data has proven to be an effective strategy for improving large language model (LLM) performance. Yet, as these models grow increasingly powerful and widely deployed, the cost of inference has become a pressing concern. Despite its importance, the trade-off between model accuracy and inference efficiency remains underexplored. In this work, we examine how key architectural factors, hidden size, the allocation of parameters between MLP and attention (mlp-to-attention ratio), and grouped-query attention (GQA), influence both inference cost and accuracy. We introduce a conditional scaling law that augments the Chinchilla framework with architectural information, along with a search framework for identifying architectures that are simultaneously inference-efficient and accurate. To validate our approach, we train more than 200 models spanning 80M to 3B parameters and 8B to 100B training tokens, and fit the proposed conditional scaling law. Our results show that the conditional scaling law reliably predicts optimal architectural choices and that the resulting models outperform existing open-source baselines. Under the same training budget, optimized architectures achieve up to 2.1% higher accuracy and 42% greater inference throughput compared to LLaMA-3.2.

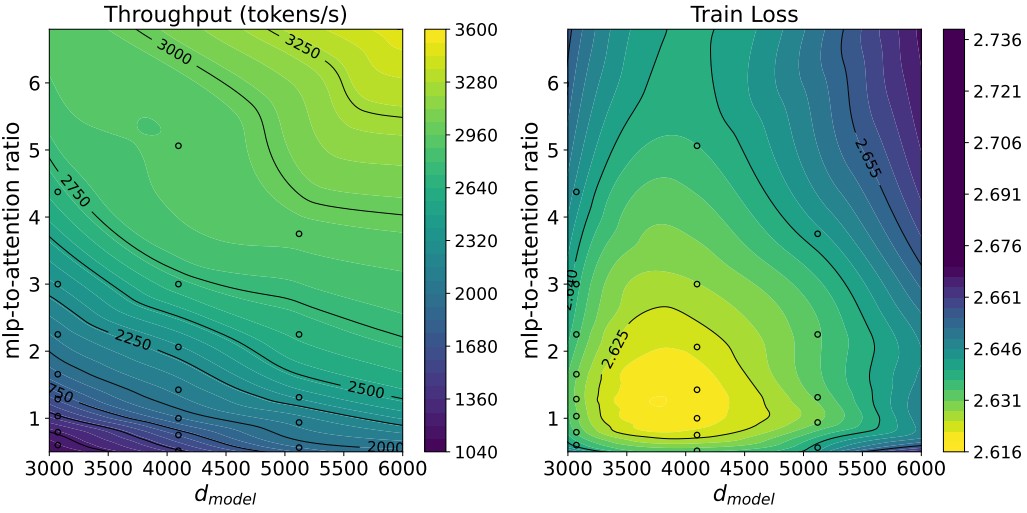

Figure 1: **Scaling sweep results.** (left) Inference throughput (tokens/s) and (right) Scaling-law-predicted training loss contours over hidden size $d_{\text{model}}$ and mlp-to-attention ratio. Our conditional scaling law enables concurrent gains in throughput and reductions in predicted training loss under a fixed parameter budget. Dotted points indicate the architectures used to fit the scaling law.

---

[*]Work done during internship at Amazon Web Services.
[†]Correspondence to: Tao Yu (taou@amazon.com).

# 1 INTRODUCTION

Scaling law studies Kaplan et al. (2020); Hoffmann et al. (2022); Muennighoff et al. (2023); Krajewski et al. (2024); Abnar et al. (2025) have shown that increasing model parameters, training tokens, dataset quality, and compute budget consistently reduces pre-training loss, improves downstream task performance Hendrycks et al. (2021); Austin et al. (2021), and enables the emergence of novel capabilities Wei et al. (2022). These insights have driven the development of many state-of-the-art large language models Touvron et al. (2023); Yang et al. (2025); Guo et al. (2025).

However, as the field advances, it has become increasingly clear that focusing exclusively on training overlooks the practical challenges of deploying these models at scale Chien et al. (2023); Wu et al. (2024); Muhamed et al. (2023). A major limitation of existing scaling laws is their omission of inference costs, which constitute the dominant expense in deploying large models in real-world applications Sardana et al. (2023); Park et al. (2024). Moreover, the growing use of LLMs in reasoning systems highlights the need for scaling laws that account for inference costs Snell et al. (2024); Brown et al. (2024); Luo et al. (2024); Qi et al. (2024); Guan et al. (2025). Therefore, we ask the following question:

> *Can we explicitly capture the trade-off between inference efficiency and accuracy of large language models?*

To address this question, a recent study Sardana et al. (2023) proposed scaling laws that incorporate the total FLOPs from both training and inference. However, their formulation requires estimating the total number of tokens generated over a model's entire lifespan. Because inference is performed repeatedly during deployment, this assumption renders the proposed scaling law impractical for real-world use. Another study Bian et al. (2025) extends Chinchilla scaling laws by incorporating model architecture. However, this work has notable limitations. First, the study considers only the aspect ratio, defined as hidden size over number of layers, as the architectural factor. Yet, as shown in Figure 2, aspect ratio alone fails to capture the full range of factors that influence inference efficiency in large language models. Second, the depth of the model strongly influences accuracy: cutting layers tends to impair the model's generalization after fine-tuning Petty et al. (2023). Finally, the study lacks a general framework for incorporating broader architectural factors, including hidden size and GQA, into scaling laws.

In this work, we fix the number of layers and study the effect of other architectural factors, including GQA, hidden size, and the mlp-to-attention ratio. This design choice is motivated by recent open-weight models such as LLaMA Touvron et al. (2023), Qwen Yang et al. (2025), Gemma Team et al. (2024a), and Phi Abdin et al. (2024), which, despite having a comparable number of parameters, adopt markedly different architectural designs.

Our primary goal is to investigate how model architecture influences both inference efficiency and model accuracy. We begin by comparing the inference efficiency of models with identical parameter counts but varying architectures. Next, we train over 200 models, ranging from 80M to 297M parameters on up to 30B tokens, to systematically characterize the relationship between architectural design and accuracy. Guided by these empirical findings, we introduce a conditional extension of the Chinchilla scaling laws that incorporates architectural parameters, establishing a general framework for identifying model architectures that balance inference efficiency and performance.

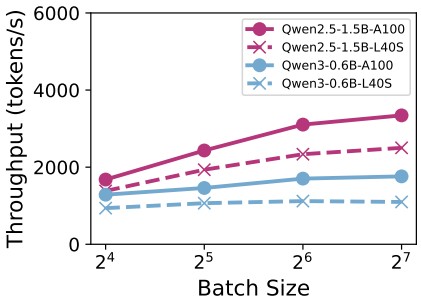

Figure 2: Although larger models generally achieve lower inference throughput than smaller ones, Qwen2.5-1.5B outperforms Qwen3-0.6B. Despite having the same number of layers, Qwen2.5-1.5B benefits from a higher hidden size, GQA, and mlp-to-attention ratio.

Finally, we validate this framework by fitting the proposed scaling law on models between 80M and 297M parameters, and evaluating its predictions when scaling up to pretrain 3B-parameter models. Our results demonstrate that, under identical training setups, the derived optimal 3B-parameter ar-

chitecture achieves up to $42\%$ higher inference throughput than the LLaMA-3.2-3B architecture, while maintaining better accuracy.

## 2 BACKGROUND

Accurately predicting the performance of large language models during scaling is essential. This enables us to answer key questions: (i) what is the optimal allocation of available resources between model size and training tokens, and (ii) what performance gains can be expected from additional resources? Fortunately, the model loss has been observed to follow a power-law relationship with respect to the number of parameters $N$ and training tokens $D$ Hoffmann et al. (2022); Muennighoff et al. (2023) with:

$$L(N, D) = E + \frac{A}{N^\alpha} + \frac{B}{D^\beta} \tag{1}$$

where $L$ is the model loss, $N$ is the number of total parameters and $D$ is the number of tokens used for training and $A$, $B$, $E$, $\alpha$, $\beta$ are parameters to be learned.

To fit the learnable parameters in Eq. (1), Chinchilla Hoffmann et al. (2022) employs two strategies: (i) training models with a fixed number of parameters while varying the number of training tokens, and (ii) training models under a fixed compute budget[1], varying both parameters and tokens. The resulting data are combined to fit the learned parameters in Eq. (1). With the fitted scaling laws, Chinchilla addresses the following question to determine optimal allocation:

$$\arg\min_{N,D} L(N, D) \text{ s.t. FLOPs}(N, D) = C \tag{2}$$

where $C$ denotes the resource constraint, $N$ the total number of parameters, and $D$ the number of training tokens.

In this paper, we do not address how to optimally allocate compute between model size and training data under a fixed compute budget. Instead, our focus is on identifying model architectures that optimize inference efficiency and accuracy under fixed parameter and token budgets. For example, given a model with 7B parameters trained on 14T tokens, we study how to design an architecture that satisfies both efficiency and accuracy requirements.

## 3 MODEL ARCHITECTURE-AWARE SCALING LAWS

### 3.1 MODEL ARCHITECTURE VARIATIONS

The architecture of a decoder-only transformer is composed of a sequence of stacked decoder blocks, each sharing the same structure to facilitate model-parallel deployment across devices. Under this design, the overall architecture of dense LLMs is primarily determined by the hidden size and the MLP intermediate size, which together specify the attention and MLP layers structure. This work studies the optimal model architecture given a fixed total number of non-embedding parameters $N_{\text{non-embed}}$ (at different levels). Although the number of layers $n_{\text{layer}}$ also plays a critical role (closely related to aspect ratio (Petty et al., 2023)), varying $n_{\text{layer}}$ under a fixed $N_{\text{non-embed}}$ substantially impacts both inference cost and accuracy (Tay et al., 2021; Alabdulmohsin et al., 2023). Therefore, we fix $n_{\text{layer}}$ and focus on the effects of hidden size $d_{\text{model}}$ and the mlp-to-attention ratio $r_{\text{mlp/attn}}$ on inference efficiency (§3.2) and accuracy (§3.3), noting that $n_{\text{layer}}$ still varies across different $N_{\text{non-embed}}$ levels. In §3.3, we introduce a conditional scaling law to predict the performance of architectural variants, and in §3.4, we present a lightweight framework for identifying architectures that optimally balance inference efficiency and accuracy.

Note that the number of attention parameters is primarily determined by the hidden size $d_{\text{model}}$ and the attention projection dimension, since most open-weight models adopt non-square $q, k, v$ projection matrices, as seen in Gemma (Team et al., 2024a) and Qwen3 (Yang et al., 2025). For consistency, we fix the per-head dimension $d_{\text{head}}$ to 64 for models with $N_{\text{non-embed}} \leq$1B and to 128

---

[1]The compute cost is approximated as FLOPs$(N, D) \approx 6ND$ in Hoffmann et al. (2022); Muennighoff et al. (2023), where $N$ denotes the number of parameters and $D$ the number of training tokens. In this work, we adopt the same settings as prior studies.

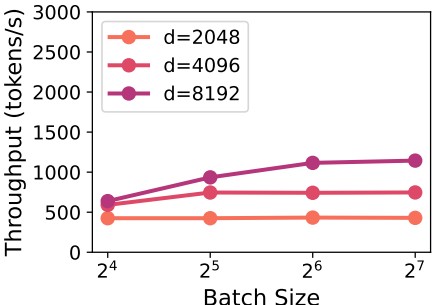 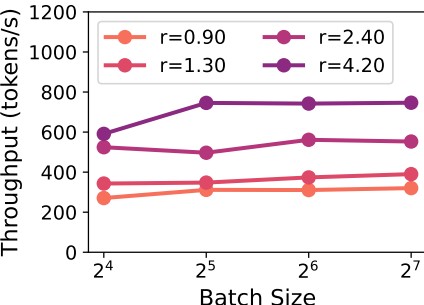

Figure 3: **Inference throughput.** (left) hidden size $d = d_{\mathrm{model}}$ and (right) mlp-to-attention ratio $r = r_{\mathrm{mlp/attn}}$ on the 8B model. Under a fixed parameter budget $N_{\mathrm{non\text{-}embed}}$, larger hidden sizes and higher mlp-to-attention ratios improve inference throughput for varying batch sizes.

for models with $N_{\mathrm{non\text{-}embed}} \geq 3$B. Consequently, to maintain a constant $r_{\mathrm{mlp/attn}}$, we adjust the number of attention heads $n_{\mathrm{head}}$ rather than altering the projection dimension directly. This design choice also provides flexibility to incorporate architectural variants such as grouped-query attention.

## 3.2 INFERENCE EFFICIENCY

Inspired by the success and widespread adoption of open-weight dense models such as Qwen3 (Yang et al., 2025), LLaMA-3.2 (Dubey et al., 2024), and the Gemma-2 (Team et al., 2024b) family, we construct architectural variants by modifying the configurations of the LLaMA-3.2 and Qwen3 dense models (Figure 12-14 in Appendix F). In addition to hidden size and the mlp-to-attention ratio, we find that group-query attention has a critical impact on inference efficiency, even though it only modestly reduces the number of attention parameters (by shrinking the key and value matrices). To disentangle these effects, we conduct controlled ablations of hidden size, MLP-to-attention ratio, and GQA under the following setups:

- *hidden size $d_{model}$*: fix $N_{\mathrm{non\text{-}embed}}$, $r_{\mathrm{mlp/attn}}$ and GQA= 4, vary $d_{\mathrm{model}}$ and number of attention heads $n_{\mathrm{head}}$ (Figure 3 left).
- *mlp-to-attention ratio $r_{mlp/attn}$*: fix $N_{\mathrm{non\text{-}embed}}$, $d_{\mathrm{model}}$ and GQA= 4, vary $n_{\mathrm{head}}$ and intermediate size (Figure 3 right).
- *GQA:* fix $N_{\mathrm{non\text{-}embed}}$, $d_{\mathrm{model}}$ and $r_{\mathrm{mlp/attn}}$, vary $n_{\mathrm{head}}$ and number of key-value heads (Appendix F).

Figure 3 shows the ablation of varying hidden sizes $d_{\mathrm{model}}$ and mlp-to-attention $r_{\mathrm{mlp/attn}}$ on the LLaMA-3.1-8B model variants. We observe that larger hidden size (or fewer attention heads) and higher mlp-to-attention ratios improve inference throughput. Similar trends are observed in the LLaMA-3.2-1B and 3B model variants (Appendix F). These gains arise in part because larger $d_{\mathrm{model}}$ and higher $r_{\mathrm{mlp/attn}}$ reduce the total FLOPs, as detailed in the inference FLOPs analysis (Appendix K). In addition, these architectural choices shrink the KV cache, lowering I/O cost during inference and further improving throughput Adnan et al. (2024). Figure 11 in Appendix F presents the GQA ablation, confirming prior observations Ainslie et al. (2023) that increasing GQA consistently improves inference throughput. A comparable set of ablation experiments on Qwen3 models, also reported in Appendix F, further corroborates these findings.

## 3.3 A CONDITIONAL SCALING LAW

Improving inference efficiency should not come at the expense of significantly reducing model accuracy, making it crucial to understand how architectural choices affect accuracy and training loss. Because training large-scale language models is prohibitively expensive, a common strategy is to study smaller models and use scaling laws to extrapolate insights to larger scales, for example, the Chinchilla scaling laws (Hoffmann et al., 2022). However, incorporating multiple architectural factors into such laws remains challenging. To address this, we examine the effect of architectural

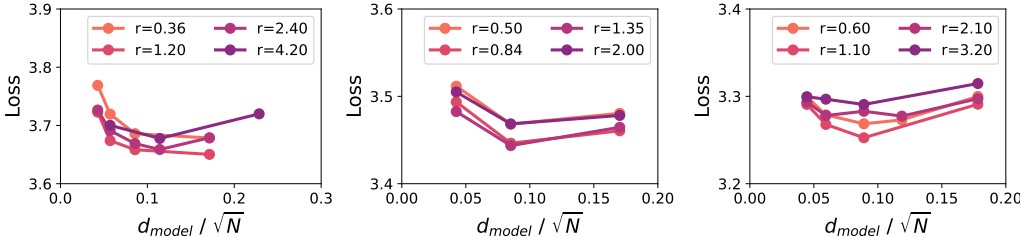

Figure 4: **Loss vs. hidden size.** (left) 80M model variants; (center) 145M model variants; (right) 297M model variants. Across model sizes, the relationship between training loss and $d_{\mathrm{model}}/\sqrt{N}$ exhibits a consistent U-shaped curve when architectural factors such as GQA and the MLP-to-attention ratio are held fixed. The legend denotes the MLP-to-attention ratio $r = r_{\mathrm{mlp/attn}}$ for each model.

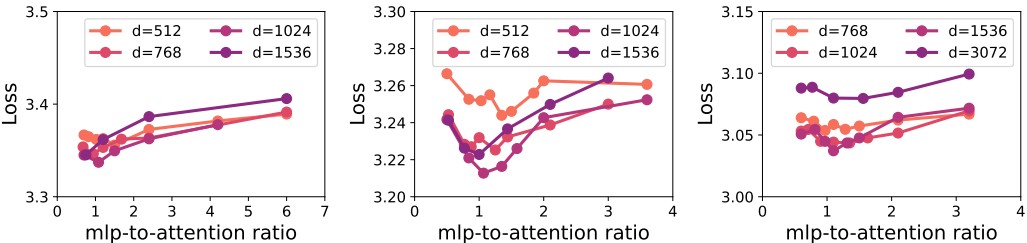

Figure 5: **Loss vs. MLP-to-attention ratio.** (left) 80M model variants; (center) 145M model variants; (right) 297M model variants. Across model sizes, the relationship between training loss and $r_{\mathrm{mlp/attn}}$ exhibits a consistent U-shaped curve when architectural factors such as GQA and hidden size are held fixed. The legend denotes the hidden size $d = d_{\mathrm{model}}$ for each model.

choices on training loss $L$ in a conditional manner, varying one factor at a time while keeping the others fixed.

**hidden size $d_{\mathbf{model}}$.** We note that $d_{\mathrm{model}}$ generally scales linearly with $\sqrt{N_{\mathrm{non\text{-}embed}}}$. Assuming squared attention weight matrices, the number of attention parameters $N_{\mathrm{attn}}$ can be expressed as

$$4d_{model}^2 \propto N_{\mathrm{attn}} = N_{\mathrm{non\text{-}embed}} \times \frac{1}{r+1},$$

where $r = r_{\mathrm{mlp/attn}}$ is fixed, and the constant factor 4 arises from the query, key, value, and output projection layers in each attention block. To capture this scaling behavior, we normalize $d_{\mathrm{model}}$ by $\sqrt{N_{\mathrm{non\text{-}embed}}}$ and examine its relation to loss $L$ in Figure 4. The resulting U-shaped curves $L(d/\sqrt{N} \mid r, N, D)$ exhibit nearly identical optima across different model sizes. Moreover, Figure 4 confirms that excessively large hidden sizes, which reduce the number of attention heads $n_{\mathrm{head}}$, can degrade accuracy—a phenomenon consistently observed in prior analyses of transformer capacity and head allocation (Kaplan et al., 2020; Hoffmann et al., 2022).

**mlp-to-attention ratio $r_{\mathbf{mlp/attn}}$.** Figure 5 illustrates how the loss varies with $r_{\mathrm{mlp/attn}}$, conditioned on $d_{\mathrm{model}}$ fixed at different levels, where we consistently observe a U-shaped curve $L(r \mid d/\sqrt{N}, N, D)$. While the attention mechanism is central to the success of transformers (Vaswani, 2017), recent open-weight models have allocated a progressively smaller fraction of parameters to attention as overall model size increases (e.g., LLaMA and Qwen families). Our analysis indicates that this trend is not universally optimal: there exists an interior optimum in the allocation of attention parameters, and deviating from it in either direction degrades model performance. This suggests that careful tuning of the mlp-to-attention ratio is critical for scaling transformers effectively.

As shown in Figures 4 and 5, both hidden size and the MLP-to-attention ratio exhibit U-shaped relationships with training loss. To capture these trends, we fit the function $c_0 + c_1 \log x + c_2/x$ separately for $x = r_{\mathrm{mlp/attn}}$ and $d_{\mathrm{model}}/\sqrt{N_{\mathrm{non\text{-}embed}}}$. This formulation effectively models the U-shaped behavior while ensuring sublinear growth as $x$ increases. However, incorporating $r_{\mathrm{mlp/attn}}$,

$d_{\text{model}}$, $N$, and $D$ into a unified, architecture-aware scaling law remains challenging. Since fitting a single all-purpose scaling law $L(d/\sqrt{N}, r, N, D)$ is unrealistic across all possible configurations, we instead propose a two-step conditional approach:

1. For given $N$ and $D$, obtain the optimal loss $L_{\text{opt}}(N, D) = \min L(N, D) = \min \left( E + \frac{A}{N^\alpha} + \frac{B}{D^\beta} \right)$ from the Chinchilla scaling law (Eq. 1) as a reference point.
2. Calibrate the loss of architectural variants $L(d/\sqrt{N}, r \mid N, D)$ relative to this reference.

We focus on two simple and transparent calibration schemes:

- (multiplicative)

$$L(d/\sqrt{N}, r \mid N, D) = (a_0 + a_1 \log(\frac{d}{\sqrt{N}}) + a_2 \frac{\sqrt{N}}{d}) \cdot (b_0 + b_1 \log r + \frac{b_2}{r}) \cdot L_{\text{opt}} \quad (3)$$

- (additive) $L(d/\sqrt{N}, r \mid N, D) = (a_0 + a_1 \log(\frac{d}{\sqrt{N}}) + a_2 \frac{\sqrt{N}}{d}) + (b_1 \log r + \frac{b_2}{r}) + L_{\text{opt}}$

Here, $a_i$ and $b_i$ are learnable parameters that are shared across all $N, D$. Note that both functional forms assume the effects of $r_{\text{mlp/attn}}$ and $d_{\text{model}}$ on loss are separable.

## 3.4 SEARCHING FOR INFERENCE-EFFICIENT ACCURATE MODELS

With the conditional scaling law, we can identify architectures that are both inference-efficient and accurate by solving the following optimization problem: given $N$, $D$, and a set of architectural choices $P$,

$$\text{argmax}_P I_N(P), \qquad \text{s.t.} \quad L(P \mid N, D) \leq L_t, \qquad (4)$$

where $I_N(P)$ denotes the inference efficiency of an architecture $P$ with total $N_{\text{non-embed}}$ parameters, and $L_t, (\geq L_{\text{opt}})$ is the maximum allowable training loss.

As shown in Figure 11 (Appendix F), GQA has a substantial impact on inference efficiency; However, unlike hidden size and the mlp-to-attention ratio, GQA does not exhibit a consistent continuous relationship with loss (Figure 24, Appendix I) and is highly variable, making it challenging to identify settings that achieve both accuracy and efficiency. Fortunately, the search space for GQA is relatively small once $N_{\text{non-embed}}$, $d_{\text{model}}$, and $r_{\text{mlp/attn}}$ are fixed, since GQA must be a prime factor of the number of attention heads $n_{\text{head}}$. In practice, we perform a local GQA search by enumerating feasible values and applying early stopping once performance falls below that of the GQA$= 4$ baseline. Algorithm 1 summarizes our overall framework for identifying inference-efficient and accurate architectures.

---

**Algorithm 1:** Searching for Inference-Efficient Accurate Model

---

**Input:** Model parameters $N$, training tokens $D$, target loss $L_t$; inference efficiency $I_N(\cdot)$;
        optional: the optimal loss $L_{\text{opt}}(N, D)$

Train smaller models to fit the Chinchilla scaling laws (Eq. 1) if $L_{\text{opt}}(N, D)$ is unavailable
Solve the constrained optimization (Eq. 4) for $d_{\text{model}}$, $r_{\text{mlp/attn}}$ and corresponding architecture $P$
Perform a local search over GQA values with early stopping to maximize inference efficiency
**return** Final model architecture $\{P, \text{GQA}\}$

---

## 4 EXPERIMENT SETUP

We first detail the experimental setup of training, inference, and downstream task evaluation, and then describe how we derive the conditional scaling law and scale up to larger sizes.

**Training Setup.** We sample the training data from Dolma-v1.7 Soldaini et al. (2024), which contains data from 15 different sources. Tokens are sampled with probability proportional to each source's contribution, ensuring the sampled dataset preserves a similar distribution to Dolma-v1.7. We train decoder-only LLaMA-3.2 (Dubey et al., 2024) style transformers with $N_{\text{non-embed}}$ in $\{80M, 145M, 297M, 1B, 3B\}$, for each $N_{\text{non-embed}}$, we obtain model architecture candidates by varying hidden size $d_{\text{model}}/\sqrt{N_{\text{non-embed}}}$ and mlp-to-attention ratio $r_{\text{mlp/attn}}$. (changing intermediate size

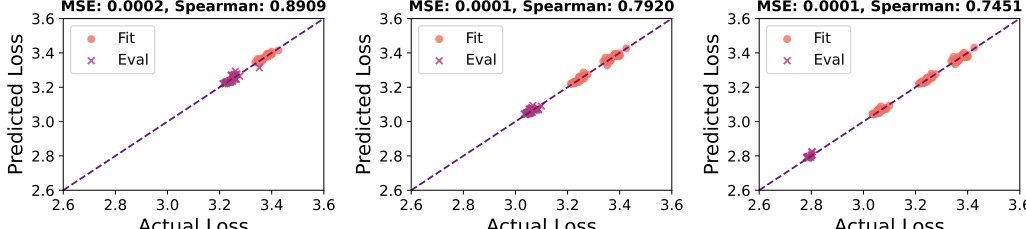

Figure 6: **Predictive performances** of the fitted conditional scaling law on: (left) Task 1: Fit on 80M, evaluate on 145M; (center) Task 2: Fit on 80, 145M, evaluate on 297M; (right) Task 3: Fit on 80, 145, 297M, evaluate on 1B. Orange dots denote fitting data points, and purple crosses indicate the test data points. We compare scaling-law predicted loss with actual pretraining loss of architectures and observed a consistently low MSE and high Spearman correlation across model scales.

and number of attention heads $n_{\text{head}}$) while holding other architectural factors fixed e.g. GQA= 4. A full list of over 200 model architectures used can be found in Appendix D. All models are trained on $100N_{\text{non-emb}}$ tokens ($5\times$ Chinchilla optimal) to ensure convergence. We tuned training hyper-parameters (mainly following prior work Chen et al. (2025)), with a full list in Appendix E.

**Inference Setup.** We evaluate the inference efficiency using the vLLM framework Kwon et al. (2023). By default, inputs consist of 4096 tokens and outputs of 1024 tokens. We report the av-eraged inference throughput (tokens/second) from 5 repeated runs. Unless otherwise specified, all experiments are conducted on NVIDIA Ampere A100 GPUs (40GB) with vLLM.

**LLM Evaluation Setup.** Following prior works Biderman et al. (2023); Zhang et al. (2024), we evaluate pretrained models in the zero-shot setting using `lm-evaluation-harness`[2] on nine benchmarks: ARC-Easy Clark et al. (2018), ARC-Challenge Clark et al. (2018), LAM-BADA Paperno et al. (2016), HellaSwag Zellers et al. (2019), OpenBookQA Mihaylov et al. (2018), PIQA Bisk et al. (2020), SciQ Welbl et al. (2017), WinoGrande Sakaguchi et al. (2021), and CoQA Reddy et al. (2019).

**Fitting Scaling Laws.** Following Gadre et al. (2024); Bian et al. (2025), we use the Levenberg-Marquardt algorithm to fit the conditional scaling laws (Eq. 3). The Levenberg–Marquardt algorithm does least-squares curve fitting by estimating $\hat{\beta}$ as the solution to $\arg\min_{\beta} \sum_{i=1}^{m} [y_i - f(x_i, \beta)]^2$, where $(x_i, y_i)$ are the observed data pairs. Note that instead of fitting the Chinchilla scaling law, we empirically searched over architecture variants to find the optimal loss $L_{\text{opt}}(N, D)$ for $N_{\text{non-embed}} < 1$B scale.

We scale up the scale law fitting in the following progressive manner:

(Task 1) fit on the 80M results and evaluate on 145M results;
(Task 2) fit on 80, 145M results and evaluate on 297M results;
(Task 3) fit on 80, 145, 297M results and evaluate on 1B results;

This ensures a robust and consistent way of scaling up the model sizes and evaluating our conditional scaling law. Following prior work Kumar et al. (2024), we evaluate the fitted scaling law with mean squared error (MSE) metric, defined as $\frac{1}{n} \sum_{i=1}^{n} (l_i - \hat{l}_i)^2$ where $l_i$ denotes the actual loss and $\hat{l}_i$ the predicted loss. We additionally report the Spearman's rank correlation coefficient Spearman (1961) to compare predicted and actual rankings. Both metrics are calculated on the val data points.

## 5   EXPERIMENT RESULTS

We begin by evaluating the predictive performances of the conditional scaling laws with multiplica-tive calibration. We then conduct ablation studies to assess the impact of data selection and to

---

[2]`https://github.com/EleutherAI/lm-evaluation-harness`

Table 1: **Large-Scale Model Results.** We evaluate the scaling laws at 1B and 3B scales by training Panda-1B, Surefire-1B, and Panda-3B, and compare them with LLaMA-3.2-1B and LLaMA-3.2-3B, respectively. The Avg. column reports the mean accuracy across the nine downstream tasks. Panda-1B and 3B are trained using the optimal architectural configurations predicted by our scaling laws, whereas Surefire-1B and 3B satisfy the loss constraint in Eq. (4) and achieve Pareto optimality.

| Models | $d_{\text{model}}$ | $f_{\text{size}}$ | $n_{\text{layers}}$ | GQA | $d_{\text{model}}/\sqrt{N}$ | $r$ | Loss ($\downarrow$) | Avg. ($\uparrow$) |
|---|---|---|---|---|---|---|---|---|
| LLaMA-3.2-1B | 2048 | 8192 | 16 | 4 | 0.066 | 4.80 | 2.803 | 54.9 |
| Panda-1B | 2560 | 4096 | 16 | 4 | 0.082 | 1.07 | 2.782 | 57.0 |
| Surefire-1B | 2560 | 6144 | 16 | 9 | 0.082 | 3.60 | 2.804 | 55.4 |
| LLaMA-3.2-3B | 3072 | 8192 | 28 | 3 | 0.058 | 3 | 2.625 | 61.9 |
| Panda-3B | 4096 | 4096 | 28 | 3 | 0.077 | 1 | 2.619 | 62.5 |
| Surefire-3B | 4096 | 4096 | 28 | 7 | 0.077 | 1 | 2.620 | 62.6 |

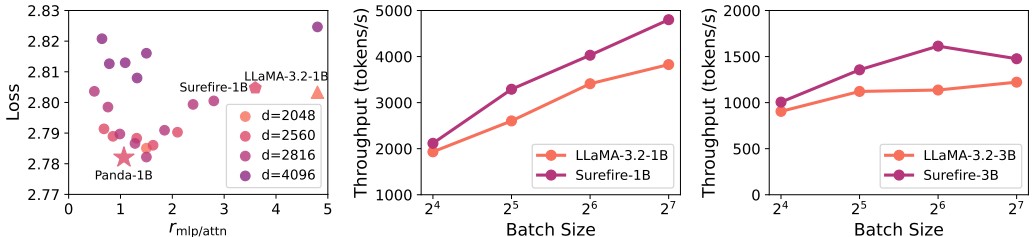

Figure 7: **Results for 1B and 3B models.** (left) Panda-1B closely follows the scaling law predictions for minimizing training loss. (center & right) Inference throughput comparison between LLaMA-3.2 and Surefire models, where Surefire is consistently efficient across all batch sizes.

evaluate the performance of the scaling laws under additive calibration. Finally, we apply the fitted scaling laws to guide the training of large-scale models following the search framework (§5.1).

**Predictive Accuracy.** As Task 1-3 described in §4, we fit the conditional scaling laws on 80M, (80M, 145M), and (80M, 145M, 297M) loss-architecture data points, and subsequently evaluate on 145M, 297M, and 1B data, respectively. In Figure 6, the low MSE and high Spearman correlation in tasks across different model scales validate the effectiveness and strong predictive performance of the proposed conditional scaling laws.

**Ablation of Outliers.** The mlp-to-attention ratio $r_{\text{mlp/attn}}$ of open-weights models typically fall between $0.5$ and $5$, for example, the mlp-to-attention ratio for LLaMA-3.2-1B, LLaMA-3.2-3B, Qwen3-0.6B, and Qwen3-8B are 4.8, 3.0, 1.5 and 3.6, respectively. In Figure 6, we fit the conditional scaling law using only model architectures with $r_{\text{mlp/attn}} \in [0.5, 5]$. We ablate this choice by training model architectures with outlier $r_{\text{mlp/attn}}$ below $0.5$ and above $5$ (such as $0.1, 12.6$) in Appendix D. In Figure 25 (left) and Figure 25 (center) in Appendix J, we show on Task 3 a comparison of fitting the conditional scaling law without and with these outliers (with a clear Spearman correlation score degradation), which suggests to exclude extreme outliers for better predicted performances.

**Ablation of Calibration.** In Figure 25 (right), we ablate an alternative formulation of the scaling laws with additive calibration, as discussed in §3.3. The results on Task 3 show that multiplicative and additive calibrations achieve similar MSE and Spearman correlations. Note that, unlike the conventional unified formulation, both calibrations assume that the effects of $r_{\text{mlp/attn}}$ and $d_{\text{model}}$ on loss are separable. We further ablate more complex joint, non-separable formulations in Appendix J and find that they do not provide superior predictive performance. The two-step reference-and-calibration framework appears robust enough that simple calibrations perform well.

Table 2: **3B Model Ablations.** We assess the robustness of fitting-data strategy at 3B scale by training Panda-3B (using 80M, 145M, and 297M data) and Panda-3B° (using only on 1B data), and compare both with LLaMA-3.2-3B. Avg. denotes mean accuracy across nine downstream tasks.

| Models | $d_{\mathrm{model}}$ | $f_{\mathrm{size}}$ | $n_{\mathrm{layers}}$ | GQA | $d_{\mathrm{model}}/\sqrt{N}$ | $r$ | Loss ($\downarrow$) | Avg. ($\uparrow$) |
|---|---|---|---|---|---|---|---|---|
| LLaMA-3.2-3B | 3072 | 8192 | 28 | 3 | 0.058 | 4.80 | 2.625 | 61.9 |
| Panda-3B | 4096 | 4096 | 28 | 3 | 0.077 | 1 | 2.619 | 62.5 |
| Panda-3B° | 4096 | 4608 | 28 | 3 | 0.076 | 1.23 | 2.606 | 62.5 |

### 5.1 Optimal Model Architecture

**Validating the conditional scaling law.** We validate the conditional scaling law at the 1B scale by applying multiplicative calibration on Task 3 using data from the (80M, 145M, and 297M) model variants. The learned parameters are

$$a_0 = 2.697, a_1 = 0.0974, a_2 = 0.0078, b_0 = 0.3870, b_1 = 0.0063, \text{ and } b_2 = 0.0065.$$

From this, we obtain the optimal architectural configuration of $d_{\mathrm{model}}/\sqrt{N} = 0.08, r = 1.032$ for 1B model by solving $\frac{\partial L}{\partial d_{\mathrm{model}}} = 0$ and $\frac{\partial L}{\partial r} = 0$. Using this configuration, we train a LLaMA-3.2-style 1B dense model on 100B tokens, denoted as Panda-1B. Panda-1B outperforms the open-weight LLaMA-3.2-1B baseline configs by 2.1% on average across downstream tasks (Table 1). Figure 7 (left) further confirms the effectiveness of the conditional scaling law by showing that Panda-1B achieves the lowest training loss among the exhaustively trained 1B variants under the same setup.

We also scale up our methodology to 3B models. Using the same approach but with data from the 80M, 145M, 297M, and 1B variants, we fit the scaling law and obtain $d_{\mathrm{model}}/\sqrt{N} = 0.08$ and $r = 1.055$ for the Panda 3B model. Trained on 100B tokens, Panda-3B outperforms the open weight LLaMA-3.2-3B configuration by 0.6% on average across downstream tasks (Table 1).

With all components in place, we apply the search framework for inference-efficient and accurate models (Alg. 1). For the $N_{\mathrm{non\text{-}embed}} = 1\mathrm{B}$ and 3B setting trained on 100B tokens, we set the target loss $L_t$ to match the training loss achieved by the LLaMA-3.2-1B and LLaMA-3.2-3B architectures, respectively.

**Ablation of inference efficiency.** Although inference efficiency $I_N(P)$ could, in principle, be expressed analytically, it depends heavily on hardware and inference configurations. Therefore, rather than solving for $I_N(P)$ directly, we search over feasible configurations $P_i$ that satisfy the loss constraint on A100 with vLLM and select Pareto-optimal points, which we denote as Surefire-1B and Surefire-3B. Surefire-1B and Surefire-3B outperform LLaMA-3.2-1B and LLaMA-3.2-3B on downstream tasks (Table 1 with details in Appendix L) and deliver up to 42% higher inference throughput (Figure 7, center and right). We also ablate inference efficiency using both vLLM and SGLang Zheng et al. (2023) on A100 and NVIDIA H200 GPUs (Appendix F, G). The results remain consistent with our vLLM–A100 evaluation: Surefire-1B and 3B outperform LLaMA-3.2-1B and 3B across all settings, achieving up to 47% higher throughput with SGLang on H200. This demonstrates that the efficiency gains transfer across serving stacks and hardware platforms. Detailed throughput statistics are provided in Table 6.

**Ablation of fitting data strategy.** While we adopt a progressive strategy for selecting fitting data across tasks (§4), results from small models (e.g., 80M) may not reliably predict behaviors at larger scales such as 3B. To assess this, we fit the conditional scaling law for the 3B model using only the 1B variants. As shown in Figure 8, fitting with 1B data yields lower MSE and higher Spearman correlation when predicting 3B behavior, suggesting that the law's coefficients shift with model size. We therefore refit the law with multiplicative calibration using only the 1B variants, yielding the coefficients $a_0 = 2.319, a_1 = 0.238, a_2 = 0.0176, b_0 = 0.5104, b_1 = 0.0051$, and $b_2 = 0.0062$.

This produces an alternative optimal configuration for the 3B model, with $d_{\mathrm{model}}/\sqrt{N} = 0.074$ and $r = 1.229$. We train a 3B model (Panda-3B°) under this configuration on 100B tokens and compare it with both LLaMA-3.2-3B and Panda-3B (fitted from 80M, 145M, 297M, and 1B data). As shown in Table 2, Panda-3B° achieves a lower training loss and comparable downstream accuracy to Panda-

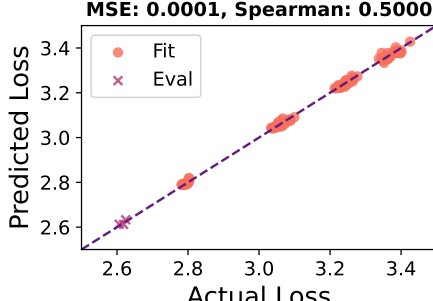 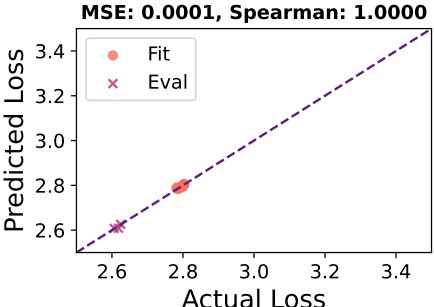

Figure 8: **Effect of the Fitting Data Strategy on Predictive Performance.** (left) Fit on 80M, 145M, 297M, 1B, evaluate on 3B; (right) Fit on 1B, evaluate on 3B. Orange dots denote fitting data, and purple crosses indicate the test data. We compare scaling-law predicted loss with actual pretraining loss of architectures and we observe that fitting the scaling laws with only 1B model data yields lower MSE and higher Spearman correlation for the 3B model loss prediction.

3B, with detailed results given in Appendix L. These findings suggest that when scaling up, it is often sufficient, and sometimes preferable, to fit the law using models within a closer size range to the target, such as about one third of its scale.

## 6 RELATED WORK

Scaling laws are powerful tools to predict the performance of large language models. Existing scaling laws Hoffmann et al. (2022); Muennighoff et al. (2023); Sardana et al. (2023); Kumar et al. (2024); Gadre et al. (2024); Ruan et al. (2024) characterize how model performance varies with model size, dataset size, data quality, and compute budget. With the rise of Mixture-of-Experts (MoE) Shazeer et al. (2017); Guo et al. (2025), a powerful architecture for large language models, recent studies Krajewski et al. (2024); Abnar et al. (2025) extend scaling laws to account for the number of experts, expert granularity, active parameters, and sparsity. Due to space constraints, we defer additional related work to Appendix B.

## 7 LIMITATIONS AND FUTURE WORK

While our team has made notable progress, several open challenges remain that offer promising directions for future research. First, due to limitations in resources and time, our evaluation does not extend to 7B models. Second, our analysis is restricted to dense models, and it remains unclear whether the results extend to Mixture of Experts (MoE) architectures Shazeer et al. (2017). While we report inference efficiency measurements for MoE models under varying architectural choices in Appendix M, we have not yet established scaling laws for MoE architectures. Finally, our analysis is limited to pre-training, and it remains unclear how the results would change under post-training.

## 8 CONCLUSION

This work explores the trade-off between model accuracy and inference cost under a fixed training budget. We begin by demonstrating how architectural choices influence both inference throughput and model accuracy. Building on this, we extend Chinchilla scaling laws to incorporate architectural factors and propose a two-step conditional framework for optimal architecture search: (i) train small models to fit the conditional scaling law (Eq. 3), and (ii) solve Eq. 4 for the predicted optimal architecture, followed by a local search over GQA to maximize inference efficiency. Using the fitted scaling laws and our framework, we trained models up to 3B parameters, achieving up to 42% higher inference throughput and 2.1% accuracy gains across nine downstream tasks. In Table 7 and Table 8 of Appendix H, we compare design choices across existing open-source models at the 1B and 3B scales, further underscoring the need for our inference-efficient accurate model designs.

REPRODUCIBILITY STATEMENT

All experiments in this work were conducted using publicly available frameworks. Section 4 provides details of our training, inference, and evaluation setups. In particular, we used `Megatron-LM` (Shoeybi et al., 2019) for model training, `vLLM` (Kwon et al., 2023) and `SGLang` (Zheng et al., 2023) for efficient inference, and `lm-eval-harness` (Gao et al., 2024a) for standardized evaluations.

ACKNOWLEDGEMENTS

Song Bian and Shivaram Venkataraman acknowledge the support of the NSF Diamond project OAC-2311767 (Democratizing Large Neural Network Model Training for Science).

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

## A LLM USAGE

We used an LLM to improve the writing by correcting grammar in our draft. It was not used to generate research ideas.

## B ADDITIONAL RELATED WORK

**Large Language Models.** Transformers Vaswani (2017) have shown strong performance across diverse downstream tasks, such as text classification Wang (2018); Sarlin et al. (2020), mathematical reasoning Cobbe et al. (2021); Hendrycks et al. (2021), and code generation Chen et al. (2021); Austin et al. (2021); Jain et al. (2024). The Transformer architecture serves as the foundation for many leading large language models, including GPT Brown et al. (2020); Achiam et al. (2023), LLaMA Touvron et al. (2023), Gemma Team et al. (2024a), Qwen Yang et al. (2025), Kimi Team et al. (2025), and DeepSeek Liu et al. (2024a); Guo et al. (2025).

**Serving Systems.** Due to the increased inference cost, many inference systems have been developed to speed up model serving Yu et al. (2022); Kwon et al. (2023); Zheng et al. (2023); Ye et al. (2025). Specifically, vLLM Kwon et al. (2023) proposes PagedAttention to manage KV cache memory more effectively, thereby improving throughput. Similarly, SGLang Zheng et al. (2023) introduces RadixAttention to achieve higher throughput and lower latency.

**Inference-Efficient Model Design.** Efforts to improve the inference efficiency of large language models generally fall into two categories: one line of work investigates the trade-offs across different model configurations Alabdulmohsin et al. (2023); Bian et al. (2025); Bian (2026), while the other focuses on designing more efficient model architectures Xiao et al. (2023); Gu & Dao (2023); Gao et al. (2024b); Jiang et al. (2024); Liu et al. (2024b); Dao & Gu (2024); Xiao et al. (2024); Yuan et al. (2025); Chandrasegaran et al. (2025).

## C OPEN-WEIGHTED MODEL ARCHITECTURES

Table 3 presents an overview of the open-weight model architectures utilized in this paper.

Table 3: **Open-Weighted Model Architectures.** We list the architectural configurations of all models used in this paper. $n_{\text{layers}}$ is the number of layers, $d_{\text{model}}$ is the hidden size, $n_{\text{heads}}$ is the number of attention heads, and $f_{\text{size}}$ is the intermediate size.

| Model Name | $n_{\text{layers}}$ | $d_{\text{model}}$ | $n_{\text{heads}}$ | $f_{\text{size}}$ | GQA |
|---|---|---|---|---|---|
| Qwen2.5-1.5B | 28 | 1536 | 12 | 8960 | 6 |
| Qwen3-0.6B | 28 | 1024 | 16 | 3072 | 2 |

## D MODEL ARCHITECTURES

Table 4 provides an overview of the model architectures, all configured with GQA = 4 and employing LLaMA-3.2 as the tokenizer.

Table 4: **Model Architectures.** We list the architectural configurations of all models trained in this paper. $N_{\text{non-embed}}$ is the total number of non-embedding parameters, $n_{\text{layers}}$ is the number of layers, $d_{\text{model}}$ is the hidden size, $n_{\text{heads}}$ is the number of attention heads, $f_{\text{size}}$ is the intermediate size, and $r_{\text{mlp/attn}}$ is the MLP-to-attention ratio.

| $N_{\text{non-embed}}$ | Variant | $n_{\text{layers}}$ | $d_{\text{model}}$ | $n_{\text{heads}}$ | $f_{\text{size}}$ | $d_{\text{model}}/\sqrt{N}$ | $r_{\text{mlp/attn}}$ |
|---|---|---|---|---|---|---|---|
| 80M | v1 | 12 | 768 | 16 | 2048 | 0.086 | 2.40 |
| 80M | v2 | 12 | 768 | 4 | 2688 | 0.086 | 12.6 |

| $N_{\text{non-embed}}$ | Variant | $n_{\text{layers}}$ | $d_{\text{model}}$ | $n_{\text{heads}}$ | $f_{\text{size}}$ | $d_{\text{model}}/\sqrt{N}$ | $r_{\text{mlp/attn}}$ |
|---|---|---|---|---|---|---|---|
| 80M | v3 | 12 | 768 | 8 | 2560 | 0.085 | 6.00 |
| 80M | v4 | 12 | 768 | 24 | 1536 | 0.087 | 1.20 |
| 80M | v5 | 12 | 768 | 32 | 1152 | 0.086 | 0.68 |
| 80M | v6 | 12 | 768 | 40 | 768 | 0.086 | 0.36 |
| 80M | v7 | 12 | 768 | 48 | 256 | 0.087 | 0.10 |
| 80M | v8 | 12 | 384 | 32 | 4096 | 0.043 | 2.40 |
| 80M | v9 | 12 | 384 | 8 | 5376 | 0.043 | 12.6 |
| 80M | v10 | 12 | 384 | 16 | 5120 | 0.042 | 6.00 |
| 80M | v11 | 12 | 384 | 48 | 3072 | 0.044 | 1.20 |
| 80M | v12 | 12 | 384 | 64 | 2304 | 0.043 | 0.68 |
| 80M | v13 | 12 | 384 | 80 | 1536 | 0.043 | 0.36 |
| 80M | v14 | 12 | 384 | 96 | 512 | 0.044 | 0.10 |
| 80M | v15 | 12 | 1536 | 8 | 1024 | 0.171 | 2.40 |
| 80M | v16 | 12 | 1536 | 4 | 1280 | 0.169 | 6.00 |
| 80M | v17 | 12 | 1536 | 12 | 768 | 0.174 | 1.20 |
| 80M | v18 | 12 | 1536 | 16 | 640 | 0.169 | 0.75 |
| 80M | v19 | 12 | 1536 | 20 | 384 | 0.171 | 0.36 |
| 80M | v20 | 12 | 1536 | 24 | 128 | 0.174 | 0.10 |
| 80M | v21 | 12 | 512 | 24 | 3072 | 0.057 | 2.40 |
| 80M | v22 | 12 | 512 | 12 | 3840 | 0.056 | 6.00 |
| 80M | v23 | 12 | 512 | 16 | 3584 | 0.057 | 4.20 |
| 80M | v24 | 12 | 512 | 36 | 2304 | 0.058 | 1.20 |
| 80M | v25 | 12 | 512 | 48 | 1792 | 0.057 | 0.70 |
| 80M | v26 | 12 | 512 | 60 | 1152 | 0.057 | 0.36 |
| 80M | v27 | 12 | 512 | 72 | 384 | 0.058 | 0.10 |
| 80M | v28 | 12 | 1024 | 12 | 1536 | 0.114 | 2.40 |
| 80M | v29 | 12 | 1024 | 8 | 1792 | 0.113 | 4.20 |
| 80M | v30 | 12 | 1024 | 16 | 1280 | 0.115 | 1.50 |
| 80M | v31 | 12 | 1024 | 24 | 896 | 0.114 | 0.70 |
| 80M | v32 | 12 | 1024 | 36 | 256 | 0.114 | 0.13 |
| 80M | v33 | 12 | 2048 | 4 | 896 | 0.226 | 4.20 |
| 80M | v34 | 12 | 2048 | 8 | 640 | 0.231 | 1.50 |
| 80M | v35 | 12 | 2048 | 16 | 256 | 0.226 | 0.30 |
| 80M | v48 | 12 | 768 | 20 | 1792 | 0.086 | 1.68 |
| 80M | v49 | 12 | 768 | 28 | 1408 | 0.086 | 0.94 |
| 80M | v50 | 12 | 384 | 40 | 3584 | 0.043 | 1.68 |
| 80M | v51 | 12 | 384 | 52 | 3072 | 0.043 | 1.11 |
| 80M | v52 | 12 | 384 | 56 | 2816 | 0.043 | 0.94 |
| 80M | v53 | 12 | 384 | 60 | 2560 | 0.043 | 0.80 |
| 80M | v54 | 12 | 512 | 32 | 2560 | 0.058 | 1.50 |
| 80M | v55 | 12 | 512 | 40 | 2176 | 0.057 | 1.02 |
| 80M | v56 | 12 | 512 | 44 | 1920 | 0.058 | 0.82 |
| 80M | v57 | 12 | 1024 | 20 | 1152 | 0.113 | 1.08 |
| 145M | v1 | 12 | 1024 | 16 | 3072 | 0.085 | 3.60 |
| 145M | v2 | 12 | 1024 | 8 | 3584 | 0.084 | 8.40 |
| 145M | v3 | 12 | 1024 | 24 | 2560 | 0.086 | 2.00 |
| 145M | v4 | 12 | 1024 | 32 | 2304 | 0.084 | 1.35 |
| 145M | v5 | 12 | 1024 | 40 | 1792 | 0.085 | 0.84 |
| 145M | v6 | 12 | 1024 | 48 | 1280 | 0.086 | 0.50 |
| 145M | v7 | 12 | 1024 | 64 | 512 | 0.085 | 0.15 |
| 145M | v8 | 12 | 512 | 32 | 6144 | 0.043 | 3.60 |
| 145M | v9 | 12 | 512 | 16 | 7168 | 0.042 | 8.40 |
| 145M | v10 | 12 | 512 | 48 | 5120 | 0.043 | 2.00 |
| 145M | v11 | 12 | 512 | 64 | 4608 | 0.042 | 1.35 |
| 145M | v12 | 12 | 512 | 80 | 3584 | 0.043 | 0.84 |
| 145M | v13 | 12 | 512 | 96 | 2560 | 0.043 | 0.50 |
| 145M | v14 | 12 | 512 | 128 | 1024 | 0.043 | 0.15 |

| $N_{\text{non-embed}}$ | Variant | $n_{\text{layers}}$ | $d_{\text{model}}$ | $n_{\text{heads}}$ | $f_{\text{size}}$ | $d_{\text{model}}/\sqrt{N}$ | $r_{\text{mlp/attn}}$ |
|---|---|---|---|---|---|---|---|
| 145M | v15 | 12 | 2048 | 8 | 1536 | 0.170 | 3.60 |
| 145M | v16 | 12 | 2048 | 4 | 1792 | 0.168 | 8.40 |
| 145M | v17 | 12 | 2048 | 12 | 1280 | 0.172 | 2.00 |
| 145M | v18 | 12 | 2048 | 16 | 1152 | 0.168 | 1.35 |
| 145M | v19 | 12 | 2048 | 20 | 896 | 0.170 | 0.84 |
| 145M | v20 | 12 | 2048 | 24 | 640 | 0.172 | 0.50 |
| 145M | v21 | 12 | 2048 | 32 | 256 | 0.170 | 0.15 |
| 145M | v22 | 12 | 768 | 24 | 3840 | 0.065 | 3.00 |
| 145M | v23 | 12 | 768 | 32 | 3584 | 0.063 | 2.10 |
| 145M | v24 | 12 | 768 | 40 | 3072 | 0.064 | 1.44 |
| 145M | v25 | 12 | 768 | 48 | 2560 | 0.065 | 1.00 |
| 145M | v26 | 12 | 768 | 56 | 2304 | 0.063 | 0.77 |
| 145M | v27 | 12 | 768 | 64 | 1792 | 0.064 | 0.53 |
| 145M | v28 | 12 | 1536 | 12 | 1920 | 0.129 | 3.00 |
| 145M | v29 | 12 | 1536 | 16 | 1792 | 0.127 | 2.10 |
| 145M | v30 | 12 | 1536 | 20 | 1536 | 0.128 | 1.44 |
| 145M | v31 | 12 | 1536 | 24 | 1280 | 0.129 | 1.00 |
| 145M | v32 | 12 | 1536 | 28 | 1152 | 0.127 | 0.77 |
| 145M | v33 | 12 | 1536 | 32 | 896 | 0.128 | 0.53 |
| 145M | v34 | 12 | 4096 | 4 | 768 | 0.340 | 3.60 |
| 145M | v35 | 12 | 4096 | 16 | 128 | 0.340 | 0.15 |
| 145M | v48 | 12 | 1024 | 28 | 2368 | 0.086 | 1.59 |
| 145M | v49 | 12 | 1024 | 36 | 2048 | 0.085 | 1.07 |
| 145M | v50 | 12 | 512 | 52 | 5120 | 0.042 | 1.85 |
| 145M | v51 | 12 | 512 | 60 | 4800 | 0.042 | 1.50 |
| 145M | v52 | 12 | 512 | 68 | 4224 | 0.043 | 1.16 |
| 145M | v53 | 12 | 512 | 72 | 3968 | 0.043 | 1.03 |
| 145M | v54 | 12 | 768 | 44 | 2944 | 0.063 | 1.25 |
| 145M | v55 | 12 | 768 | 52 | 2432 | 0.064 | 0.88 |
| 297M | v1 | 12 | 1536 | 24 | 4096 | 0.089 | 3.20 |
| 297M | v2 | 12 | 1536 | 8 | 4864 | 0.090 | 11.4 |
| 297M | v3 | 12 | 1536 | 16 | 4608 | 0.088 | 5.40 |
| 297M | v4 | 12 | 1536 | 32 | 3584 | 0.090 | 2.10 |
| 297M | v5 | 12 | 1536 | 48 | 2816 | 0.089 | 1.10 |
| 297M | v6 | 12 | 1536 | 64 | 2048 | 0.088 | 0.60 |
| 297M | v7 | 12 | 1536 | 80 | 1024 | 0.090 | 0.24 |
| 297M | v8 | 12 | 768 | 48 | 8192 | 0.045 | 3.20 |
| 297M | v9 | 12 | 768 | 16 | 9728 | 0.045 | 11.4 |
| 297M | v10 | 12 | 768 | 32 | 9216 | 0.044 | 5.40 |
| 297M | v11 | 12 | 768 | 64 | 7168 | 0.045 | 2.10 |
| 297M | v12 | 12 | 768 | 96 | 5632 | 0.045 | 1.10 |
| 297M | v13 | 12 | 768 | 128 | 4096 | 0.044 | 0.60 |
| 297M | v14 | 12 | 768 | 160 | 2048 | 0.045 | 0.24 |
| 297M | v15 | 12 | 3072 | 12 | 2048 | 0.178 | 3.20 |
| 297M | v16 | 12 | 3072 | 4 | 2432 | 0.180 | 11.4 |
| 297M | v17 | 12 | 3072 | 8 | 2304 | 0.177 | 5.40 |
| 297M | v18 | 12 | 3072 | 16 | 1792 | 0.180 | 2.10 |
| 297M | v19 | 12 | 3072 | 24 | 1408 | 0.178 | 1.10 |
| 297M | v20 | 12 | 3072 | 32 | 1024 | 0.177 | 0.60 |
| 297M | v21 | 12 | 3072 | 40 | 512 | 0.180 | 0.24 |
| 297M | v22 | 12 | 1024 | 36 | 6144 | 0.059 | 3.20 |
| 297M | v23 | 12 | 1024 | 12 | 7296 | 0.060 | 11.4 |
| 297M | v24 | 12 | 1024 | 24 | 6912 | 0.059 | 5.40 |
| 297M | v25 | 12 | 1024 | 48 | 5376 | 0.060 | 2.10 |
| 297M | v26 | 12 | 1024 | 72 | 4224 | 0.059 | 1.10 |
| 297M | v27 | 12 | 1024 | 96 | 3072 | 0.059 | 0.60 |
| 297M | v28 | 12 | 1024 | 120 | 1536 | 0.060 | 0.24 |

| $N_{\text{non-embed}}$ | Variant | $n_{\text{layers}}$ | $d_{\text{model}}$ | $n_{\text{heads}}$ | $f_{\text{size}}$ | $d_{\text{model}}/\sqrt{N}$ | $r_{\text{mlp/attn}}$ |
|---|---|---|---|---|---|---|---|
| 297M | v29 | 12 | 2048 | 12 | 3456 | 0.118 | 5.40 |
| 297M | v30 | 12 | 2048 | 24 | 2688 | 0.120 | 2.10 |
| 297M | v31 | 12 | 2048 | 48 | 1536 | 0.118 | 0.60 |
| 297M | v32 | 12 | 2048 | 60 | 768 | 0.120 | 0.24 |
| 297M | v45 | 12 | 1536 | 40 | 3200 | 0.089 | 1.50 |
| 297M | v46 | 12 | 1536 | 44 | 3072 | 0.089 | 1.31 |
| 297M | v47 | 12 | 1536 | 52 | 2688 | 0.088 | 0.97 |
| 297M | v48 | 12 | 1536 | 56 | 2432 | 0.089 | 0.81 |
| 297M | v49 | 12 | 768 | 80 | 6400 | 0.045 | 1.50 |
| 297M | v50 | 12 | 768 | 88 | 6016 | 0.045 | 1.28 |
| 297M | v51 | 12 | 768 | 104 | 5376 | 0.044 | 0.97 |
| 297M | v52 | 12 | 768 | 112 | 4736 | 0.045 | 0.79 |
| 297M | v53 | 12 | 3072 | 20 | 1664 | 0.177 | 1.56 |
| 297M | v54 | 12 | 3072 | 28 | 1152 | 0.180 | 0.77 |
| 297M | v55 | 12 | 1024 | 56 | 4864 | 0.060 | 1.63 |
| 297M | v56 | 12 | 1024 | 64 | 4608 | 0.060 | 1.35 |
| 297M | v57 | 12 | 1024 | 80 | 3840 | 0.059 | 0.90 |
| 297M | v58 | 12 | 1024 | 88 | 3328 | 0.060 | 0.71 |
| 297M | v59 | 12 | 2048 | 32 | 2432 | 0.117 | 1.43 |
| 297M | v60 | 12 | 2048 | 36 | 2048 | 0.120 | 1.07 |
| 297M | v61 | 12 | 2048 | 40 | 1920 | 0.118 | 0.90 |
| 297M | v62 | 12 | 2048 | 44 | 1792 | 0.117 | 0.76 |
| 1B | v1 | 16 | 2048 | 32 | 8192 | 0.066 | 4.80 |
| 1B | v2 | 16 | 2048 | 72 | 5760 | 0.067 | 1.50 |
| 1B | v3 | 16 | 2816 | 92 | 2432 | 0.089 | 0.50 |
| 1B | v4 | 16 | 2816 | 76 | 3072 | 0.091 | 0.76 |
| 1B | v5 | 16 | 2816 | 68 | 3584 | 0.090 | 0.99 |
| 1B | v6 | 16 | 2816 | 60 | 4096 | 0.090 | 1.28 |
| 1B | v7 | 16 | 2816 | 56 | 4480 | 0.089 | 1.50 |
| 1B | v8 | 16 | 2816 | 24 | 6144 | 0.089 | 4.80 |
| 1B | v9 | 16 | 2816 | 48 | 4736 | 0.090 | 1.85 |
| 1B | v10 | 16 | 2816 | 40 | 5120 | 0.090 | 2.40 |
| 1B | v11 | 16 | 2816 | 36 | 5376 | 0.090 | 2.80 |
| 1B | v12 | 16 | 2560 | 64 | 4480 | 0.082 | 1.31 |
| 1B | v13 | 16 | 2560 | 72 | 4096 | 0.082 | 1.07 |
| 1B | v14 | 16 | 2560 | 80 | 3648 | 0.082 | 0.86 |
| 1B | v15 | 16 | 2560 | 56 | 4864 | 0.082 | 1.63 |
| 1B | v16 | 16 | 2560 | 88 | 3200 | 0.082 | 0.68 |
| 1B | v17 | 16 | 2560 | 48 | 5376 | 0.082 | 2.10 |

# E HYPER-PARAMETERS

Table 5 lists the detailed hyper-parameters used for training in this paper.

Table 5: **Hyper-parameters.** We show the hyper-parameters used for training in this paper.

| Model Size | 80M | 145M | 297M | 1B | 3B |
|---|---|---|---|---|---|
| Batch Size | 256 | 256 | 512 | 512 | 512 |
| Max LR | 1.5e-3 | 1.0e-3 | 8.0e-4 | 6.0e-4 | 6.0e-4 |
| Min LR | $0.1\times$ Max LR | | | | |
| Optimizer | AdamW ($\beta_1 = 0.9$, $\beta_2 = 0.95$) | | | | |
| Weight Decay | 0.1 | | | | |
| Clip Grad Norm | 1.0 | | | | |
| LR Schedule | Cosine | | | | |
| Warmup Steps | 500 | | | | |
| Sequence Length | 2048 | | | | |

## F    ADDITIONAL INFERENCE EVALUATION RESULTS OVER A100 GPUS

In this section, we present additional inference efficiency results on NVIDIA A100 GPUs. Figure 11 presents that, when parameter count, MLP-to-Attention ratio, and hidden size are fixed, increasing GQA leads to higher inference throughput, consistent with the findings of Ainslie et al. (2023). We alter model configurations of LLaMA-3.2-1B, 3B, and LLaMA-3.1-8B in Figure 11. Moreover, we use the SGLang framework Zheng et al. (2023) to benchmark the inference throughput of LLaMA-3.2-1B, LLaMA-3.2-3B, Surefire-1B, and Surefire-3B on a single A100 GPU in Figure 15.

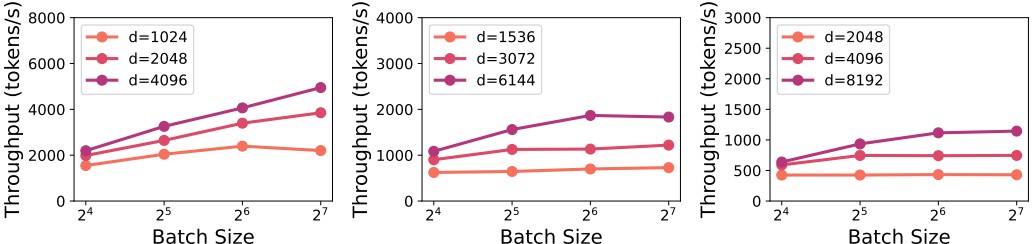

Figure 9: **Hidden size on Inference Throughput.** (left) 1B model variants; (center) 3B model variants; (right) 8B model variants. Across varying batch sizes and model scales, larger hidden sizes yield higher inference throughput under a fixed parameter budget. The legend indicates the hidden size of the models, where $d = d_{\mathrm{model}}$.

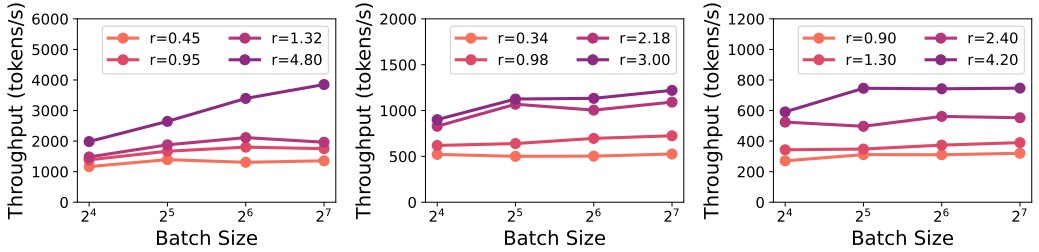

Figure 10: **MLP-to-Attention ratio on Inference Throughput.** (left) 1B model variants; (center) 3B model variants; (right) 8B model variants. Across varying batch sizes and model scales, a larger MLP-to-Attention ratio increases inference throughput under a fixed parameter budget. The legend indicates the MLP-to-Attention ratio of the models, where $r = r_{\mathrm{mlp/attn}}$.

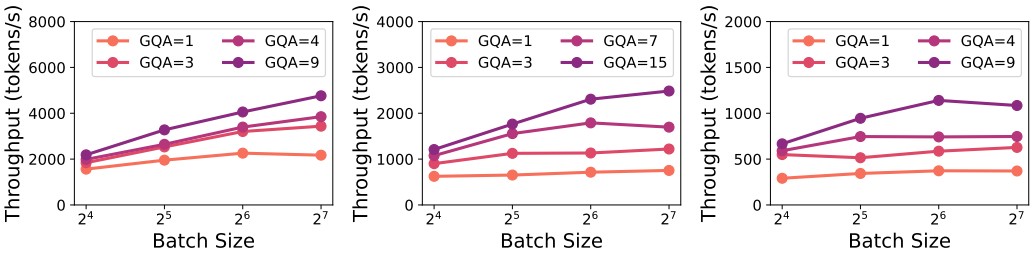

Figure 11: **GQA on Inference Throughput.** (left) 1B model variants; (center) 3B model variants; (right) 8B model variants. This figure shows the impact of GQA on inference throughput. With the total parameter count fixed, hidden size is set to 2048 (1B), 3072 (3B), and 4096 (8B), and the MLP-to-Attention ratio is 4.0, 2.67, and 4.2, respectively. Across varying batch sizes, models with larger GQA achieve higher throughput. All evaluations are performed using the vLLM framework Kwon et al. (2023) on a single NVIDIA Ampere 40GB A100 GPU with 4096 input and 1024 output tokens.

Furthermore, we derive architectural variants by altering the configurations of Qwen3-0.6B, 1.7B, and 4B to investigate the impact of model architectural factors on inference efficiency. The results are shown in Figure 12-14.

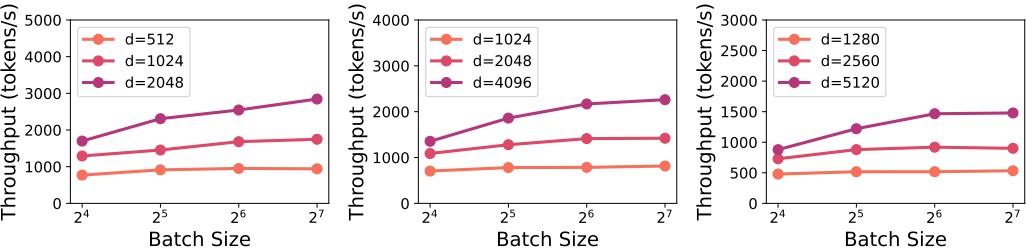

Figure 12: **Hidden size on Inference Throughput (Qwen3).** (left) Qwen3-0.6B model variants; (center) Qwen3-1.7B model variants; (right) Qwen3-4B model variants. Across varying batch sizes and model scales, larger hidden sizes yield higher inference throughput under a fixed parameter budget. The legend indicates the hidden size of the models, where $d = d_{\mathrm{model}}$. All evaluations are performed using the vLLM framework Kwon et al. (2023) on a single NVIDIA Ampere 40GB A100 GPU with 4096 input and 1024 output tokens.

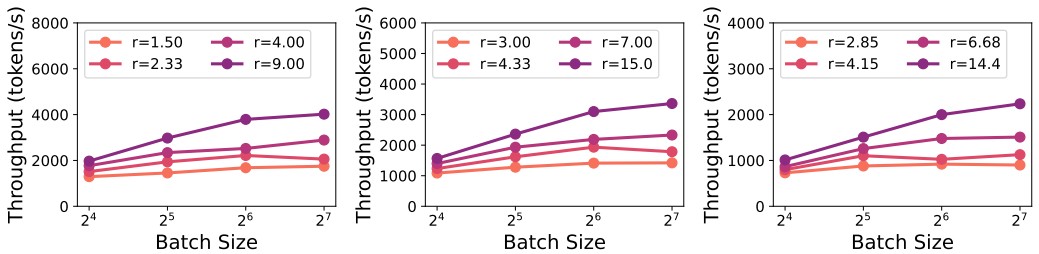

Figure 13: **MLP-to-Attention ratio on Inference Throughput (Qwen3).** (left) Qwen3-0.6B model variants; (center) Qwen3-1.7B model variants; (right) Qwen3-4B model variants. Across varying batch sizes and model scales, a larger MLP-to-Attention ratio increases inference throughput under a fixed parameter budget. The legend indicates the MLP-to-Attention ratio of the models, where $r = r_{\mathrm{mlp/attn}}$. All evaluations are performed using the vLLM framework Kwon et al. (2023) on a single NVIDIA Ampere 40GB A100 GPU with 4096 input and 1024 output tokens.

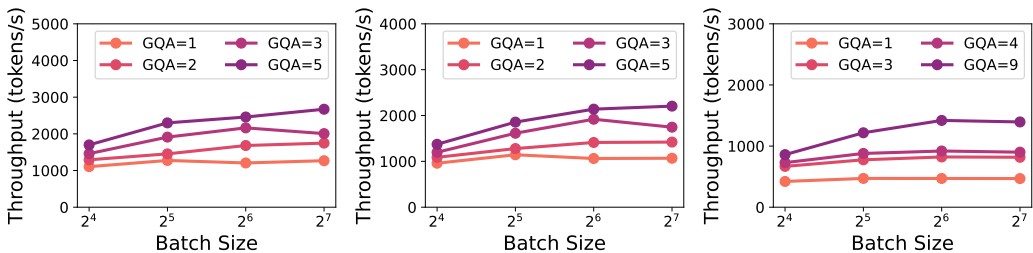

Figure 14: **GQA on Inference Throughput (Qwen3).** (left) Qwen3-0.6B model variants; (center) Qwen3-1.7B model variants; (right) Qwen3-4B model variants. This figure shows the impact of GQA on inference throughput. With the total parameter count fixed, hidden size is set to 1024 (0.6B), 2048 (1.7B), and 2560 (4B), and the MLP-to-Attention ratio is 1.5, 3.0, and 2.85, respectively. Across varying batch sizes, models with larger GQA achieve higher throughput. All evaluations are performed using the vLLM framework Kwon et al. (2023) on a single NVIDIA Ampere 40GB A100 GPU with 4096 input and 1024 output tokens.

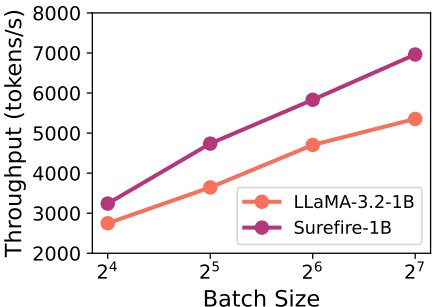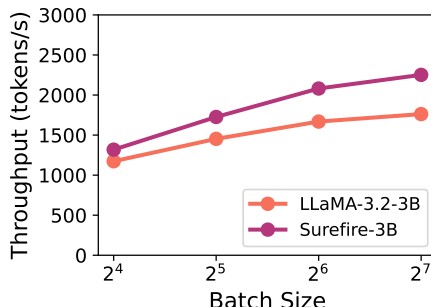

Figure 15: **Results for 1B and 3B models.** (left) Inference throughput comparison between LLaMA-3.2-1B and Surefire-1B, showing that Surefire-1B consistently achieves higher efficiency across batch sizes. (right) Inference throughput comparison between LLaMA-3.2-3B and Surefire-3B, demonstrating that Surefire-3B consistently delivers higher efficiency across all batch sizes. The results are collected using the SGLang framework Zheng et al. (2023) on a single A100 GPU with 4096 input and 1024 output tokens.

# G ADDITIONAL INFERENCE EVALUATION RESULTS OVER H200 GPUS

In this section, we present additional inference efficiency results on NVIDIA H200 GPUs. We derive architectural variants by altering the configurations of Qwen3-0.6B, 1.7B, and 4B to investigate the impact of model architectural factors on inference efficiency. The results are shown in Figure 16-18. We also compare the inference throughput of Surefire-1B, Surefire-3B, with LLaMA-3.2-1B and LLaMA-3.2-3B over H200 GPUs in Figure 19.

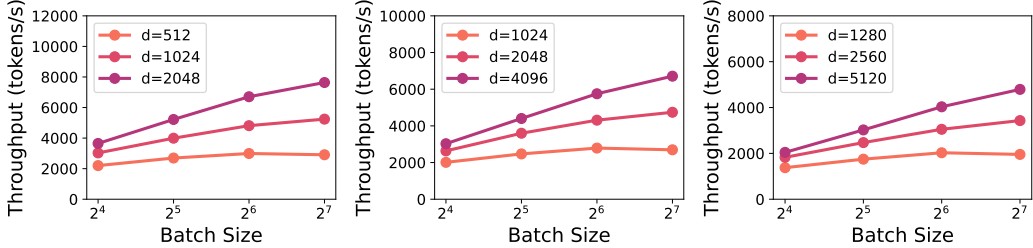

Figure 16: **Hidden size on Inference Throughput (Qwen3).** (left) Qwen3-0.6B model variants; (center) Qwen3-1.7B model variants; (right) Qwen3-4B model variants. Across varying batch sizes and model scales, larger hidden sizes yield higher inference throughput under a fixed parameter budget. The legend indicates the hidden size of the models, where $d = d_{\text{model}}$. All evaluations are performed using the vLLM framework Kwon et al. (2023) on a single NVIDIA H200 GPU with 4096 input and 1024 output tokens.

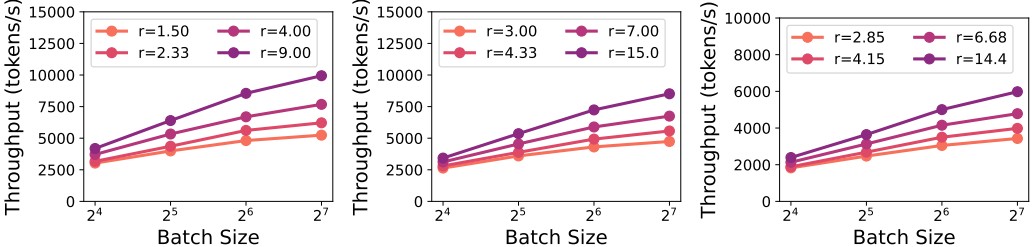

Figure 17: **MLP-to-Attention ratio on Inference Throughput (Qwen3).** (left) Qwen3-0.6B model variants; (center) Qwen3-1.7B model variants; (right) Qwen3-4B model variants. Across varying batch sizes and model scales, a larger MLP-to-Attention ratio increases inference throughput under a fixed parameter budget. The legend indicates the MLP-to-Attention ratio of the models, where $r = r_{\text{mlp/attn}}$. All evaluations are performed using the vLLM framework Kwon et al. (2023) on a single NVIDIA H200 GPU with 4096 input and 1024 output tokens.

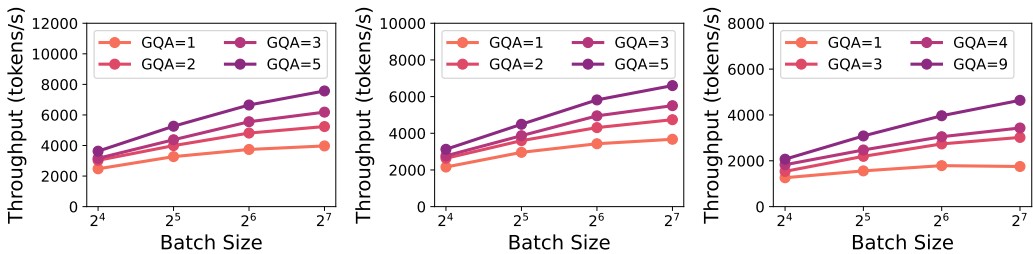

Figure 18: **GQA on Inference Throughput (Qwen3).** (left) Qwen3-0.6B model variants; (center) Qwen3-1.7B model variants; (right) Qwen3-4B model variants. This figure shows the impact of GQA on inference throughput. With the total parameter count fixed, hidden size is set to 1024 (0.6B), 2048 (1.7B), and 2560 (4B), and the MLP-to-Attention ratio is 1.5, 3.0, and 2.85, respectively. Across varying batch sizes, models with larger GQA achieve higher throughput. All evaluations are performed using the vLLM framework Kwon et al. (2023) on a single NVIDIA H200 GPU with 4096 input and 1024 output tokens.

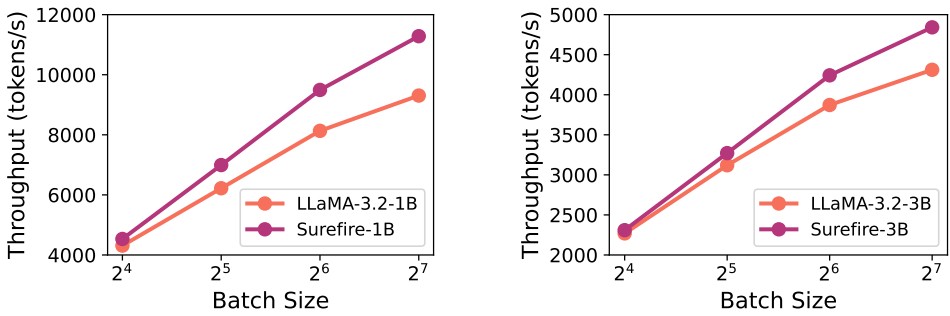

Figure 19: **Results for 1B and 3B models.** (left) Inference throughput comparison between LLaMA-3.2-1B and Surefire-1B, showing that Surefire-1B consistently achieves higher efficiency across batch sizes. (right) Inference throughput comparison between LLaMA-3.2-3B and Surefire-3B, demonstrating that Surefire-3B consistently delivers higher efficiency across all batch sizes. The results are collected using the SGLang framework Zheng et al. (2023) on a single NVIDIA H200 GPU with 4096 input and 1024 output tokens.

Furthermore, we use the SGLang framework Zheng et al. (2023) to measure the inference throughput of large language models. We construct architectural variants by modifying the configurations of

Qwen3-0.6B, 1.7B, and 4B to study how different architectural factors influence inference efficiency. The results are presented in Figure 20-22. The inference throughput of Surefire-1B and Surefire-3B compared with LLaMA-3.2-1B and LLaMA-3.2-3B on H200 GPUs is shown in Figure 23.

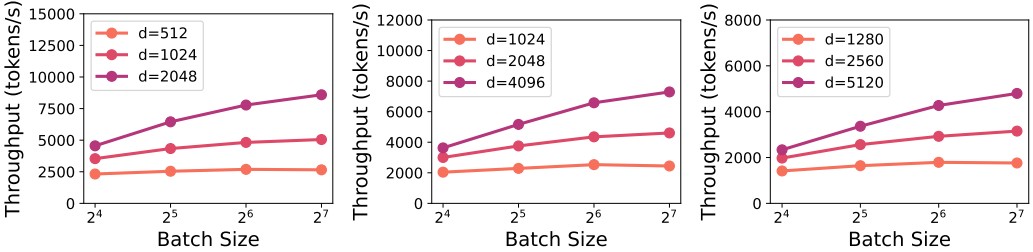

Figure 20: **Hidden size on Inference Throughput (Qwen3).** (left) Qwen3-0.6B model variants; (center) Qwen3-1.7B model variants; (right) Qwen3-4B model variants. Across varying batch sizes and model scales, larger hidden sizes yield higher inference throughput under a fixed parameter budget. The legend indicates the hidden size of the models, where $d = d_{\text{model}}$. All evaluations are performed using the SGLang framework Zheng et al. (2023) on a single NVIDIA H200 GPU with 4096 input and 1024 output tokens.

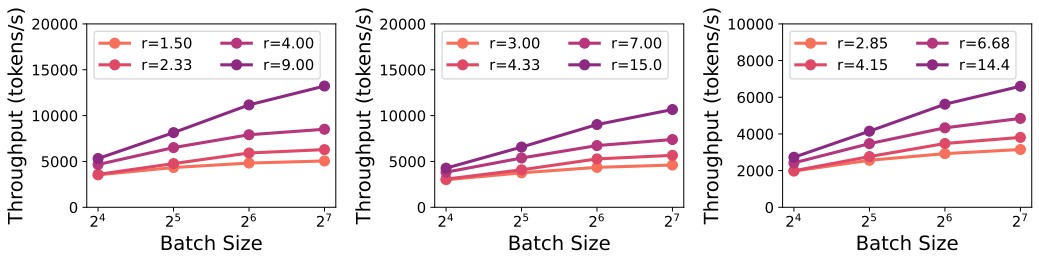

Figure 21: **MLP-to-Attention ratio on Inference Throughput (Qwen3).** (left) Qwen3-0.6B model variants; (center) Qwen3-1.7B model variants; (right) Qwen3-4B model variants. Across varying batch sizes and model scales, a larger MLP-to-Attention ratio increases inference throughput under a fixed parameter budget. The legend indicates the MLP-to-Attention ratio of the models, where $r = r_{\text{mlp/attn}}$. All evaluations are performed using the SGLang framework Zheng et al. (2023) on a single NVIDIA H200 GPU with 4096 input and 1024 output tokens.

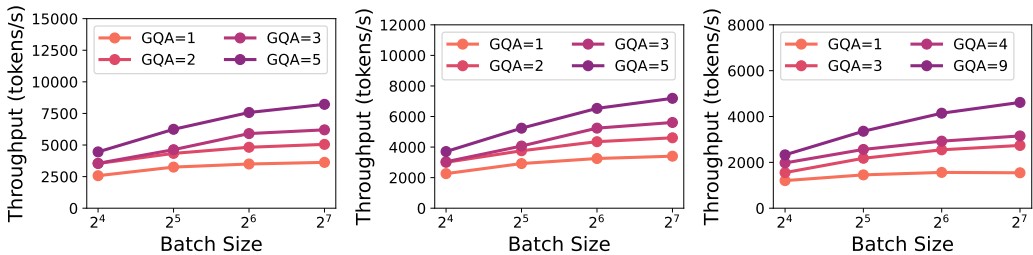

Figure 22: **GQA on Inference Throughput (Qwen3).** (left) Qwen3-0.6B model variants; (center) Qwen3-1.7B model variants; (right) Qwen3-4B model variants. This figure shows the impact of GQA on inference throughput. With the total parameter count fixed, hidden size is set to 1024 (0.6B), 2048 (1.7B), and 2560 (4B), and the MLP-to-Attention ratio is 1.5, 3.0, and 2.85, respectively. Across varying batch sizes, models with larger GQA achieve higher throughput. All evaluations are performed using the SGLang framework Zheng et al. (2023) on a single NVIDIA H200 GPU with 4096 input and 1024 output tokens.

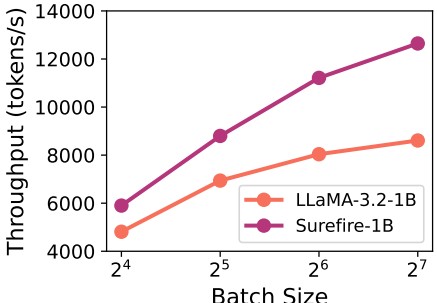 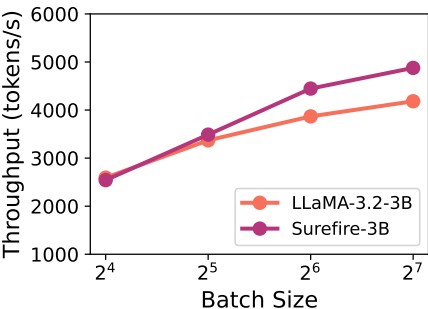

Figure 23: **Results for 1B and 3B models.** (left) Inference throughput comparison between LLaMA-3.2-1B and Surefire-1B, showing that Surefire-1B consistently achieves higher efficiency across batch sizes. (right) Inference throughput comparison between LLaMA-3.2-3B and Surefire-3B, demonstrating that Surefire-3B consistently delivers higher efficiency across all batch sizes. The results are collected using the SGLang framework Zheng et al. (2023) on a single NVIDIA H200 GPU with 4096 input and 1024 output tokens.

## H DETAILED THROUGHPUT STATISTICS

In this section, we present the detailed inference throughput results for the 1B and 3B models in Table 6.

Table 6: **Summary of Results for 1B and 3B Models.** We summarize the inference throughput (tokens/s) of LLaMA-3.2-1B, Surefire-1B, LLaMA-3.2-3B, and Surefire-3B across vLLM and SGLang on A100 and H200 GPUs using 4096 input tokens and 1024 output tokens.

| Hardware | Framework | Model | Batch Size | | | |
|---|---|---|---|---|---|---|
| | | | 16 | 32 | 64 | 128 |
| A100 | vLLM | LLaMA-3.2-1B | 1931.87 | 2602.72 | 3409.85 | 3825.91 |
| | | Surefire-1B | 2116.49 | 3290.23 | 4028.69 | 4800.05 |
| | | LLaMA-3.2-3B | 904.83 | 1121.39 | 1136.61 | 1222.03 |
| | | Surefire-3B | 1005.44 | 1356.07 | 1613.32 | 1476.22 |
| A100 | SGLang | LLaMA-3.2-1B | 2748.84 | 3643.27 | 4703.92 | 5353.29 |
| | | Surefire-1B | 3239.55 | 4737.63 | 5832.01 | 6962.24 |
| | | LLaMA-3.2-3B | 1173.51 | 1452.97 | 1668.67 | 1762.18 |
| | | Surefire-3B | 1318.23 | 1726.20 | 2081.44 | 2251.74 |
| H200 | vLLM | LLaMA-3.2-1B | 4311.97 | 6221.14 | 8131.65 | 9306.36 |
| | | Surefire-1B | 4532.85 | 6992.71 | 9493.46 | 11282.56 |
| | | LLaMA-3.2-3B | 2269.53 | 3119.94 | 3872.14 | 4311.43 |
| | | Surefire-3B | 2309.48 | 3271.63 | 4242.33 | 4841.53 |
| H200 | SGLang | LLaMA-3.2-1B | 4812.67 | 6939.88 | 8038.34 | 8608.57 |
| | | Surefire-1B | 5900.52 | 8798.68 | 11214.40 | 12645.55 |
| | | LLaMA-3.2-3B | 2593.04 | 3370.42 | 3868.42 | 4183.09 |
| | | Surefire-3B | 2542.21 | 3488.79 | 4446.66 | 4877.16 |

We further compare design choices across existing open-source models at the 1B and 3B scales in Table 7 and Table 8. For the LLaMA-3.2-1B, Panda-1B, and Surefire-1B models we pretrained, we report inference throughput (tokens/s), byte-level WikiText perplexity, and full architectural configurations in the accompanying tables. All throughput measurements are performed with vLLM on H200 GPUs using batch size 128. For the 1B scale, we include LLaMA-3.2-1B-HF and OLMo-2-1B-HF. Because OLMo supports only a 4k context window and cannot run our standard 4k/1k setup (4096 input tokens and 1024 output tokens), we additionally report results under a 2k/1k setup

(2048 input tokens and 1024 output tokens). For the 3B scale, we add LLaMA-3.2-3B-HF and Qwen2.5-3B-HF, all evaluated under the 4k/1k configuration.

Table 7: **Comparison against open-source models at the 1B scale.** We compare our pretrained LLaMA-3.2-1B, Panda-1B, and Surefire-1B models with LLaMA-3.2-1B-HF and OLMo-2-1B-HF in terms of inference throughput (on H200 GPUs using vLLM) and byte-level WikiText perplexity.

| Model | LLaMA-3.2-1B | Panda-1B | Surefire-1B | LLaMA-3.2-1B-HF | OLMo-2-1B-HF |
|---|---|---|---|---|---|
| Wikitext PPL | 1.7151 | 1.7016 | 1.7142 | 1.5807 | 1.5798 |
| Tput (4k/1k) | 9306 | 6218 | 11283 | 9306 | / |
| Tput (2k/1k) | 11948 | 8961 | 13890 | 11948 | 7486 |
| Model Architectural Config | | | | | |
| $n_{\text{layers}}$ | 16 | 16 | 16 | 16 | 16 |
| $d_{\text{model}}$ | 2048 | 2560 | 2560 | 2048 | 2048 |
| $r_{\text{mlp/attn}}$ | 4.8 | 1.067 | 3.6 | 4.8 | 3 |
| GQA | 4 | 4 | 9 | 4 | 1 |
| $N_{\text{non-embed}}$ | 973M | 975M | 965M | 973M | 1.074B |

Table 8: **Comparison against open-source models at the 3B scale.** We compare our pretrained LLaMA-3.2-3B, Panda-3B, and Surefire-3B models with LLaMA-3.2-3B-HF and Qwen2.5-3B-HF in terms of inference throughput (on H200 GPUs using vLLM) and byte-level WikiText perplexity.

| Model | LLaMA-3.2-3B | Panda-3B | Surefire-3B | LLaMA-3.2-3B-HF | Qwen2.5-3B-HF |
|---|---|---|---|---|---|
| Wikitext PPL | 1.6489 | 1.6454 | 1.6462 | 1.5164 | 1.6185 |
| Tput (4k/1k) | 4311 | 3335 | 4842 | 4311 | 6470 |
| Model Architectural Config | | | | | |
| $n_{\text{layers}}$ | 28 | 28 | 28 | 28 | 36 |
| $d_{\text{model}}$ | 3072 | 4096 | 4096 | 3072 | 2048 |
| $r_{\text{mlp/attn}}$ | 3 | 1 | 1 | 3 | 7.17 |
| GQA | 3 | 3 | 7 | 3 | 8 |
| $N_{\text{non-embed}}$ | 2.82B | 2.82B | 2.82B | 2.82B | 2.77B |

Our observations are as follows:

- OLMo-2-1B-HF is relatively close to our predicted optimal design, with an MLP-to-attention ratio of 3 (near our predicted 3.6), but remains inference-inefficient due to its hidden dimension and GQA choices.

- At the 3B scale, LLaMA-3.2-3B-HF achieves good accuracy but is not inference-efficient, while Qwen2.5-3B-HF is inference-efficient but less accurate.

These comparisons further underscore the necessity and relevance of our inference-efficient, high-accuracy model designs.

## I  ADDITIONAL RESULTS: LOSS VS. GQA

We analyze the relationship between training loss and GQA while fixing the number of parameters, hidden size, and MLP-to-Attention ratio. In order to keep the number of attention parameters fixed, we vary GQA by holding the total number of heads fixed (i.e., total heads = query-heads + KV-heads) and re-allocating this fixed budget between query and key–value heads, producing asymmetric changes in their effective dimensionalities.

As shown in Figure 24, unlike hidden size and MLP-to-Attention ratio, the relationship between loss and GQA is highly fluctuating. Varying GQA does not adjust model capacity in the coordinated way that changing $d_{\text{model}}$ or $r_{\text{mlp/attn}}$ does, where the dimensions of query, key, and value scale together predictably. Specifically, note the following facts when the total number of heads is fixed

- Increasing the number of query-heads expands the query projection dimensionality but simultaneously reduces the number of KV-heads, increasing KV sharing and thus reducing KV expressivity.
- Conversely, decreasing query-heads increases KV-head capacity (fewer replicas) but reduces the projection dimensionality of both query and KV.

These opposing effects create a tradeoff, making the relationship between GQA and training loss non-smooth and often highly fluctuating.

Prior work shows only that query and KV projections can have non-interchangeable roles (e.g., head-importance heterogeneity Voita et al. (2019)), but provides no monotonic or predictive theory for how reallocating capacity across query versus KV should affect loss. Consistent with this, recent open LLMs choose different GQA settings even within a single family: Qwen3 uses GQA = 2 for 0.6B/1.7B, GQA = 4 for 4B/8B, GQA = 5 for 14B, GQA = 8 for 32B and for the 30B-A3B MoE; LLaMA-3/3.1/3.2 likewise use GQA = 4, 8, and 3 across closely related sizes. This variation across models of similar architecture shows that GQA is treated as a discrete, model-specific hyperparameter, supporting our decision to tune it via local search rather than integrate it into the continuous scaling law.

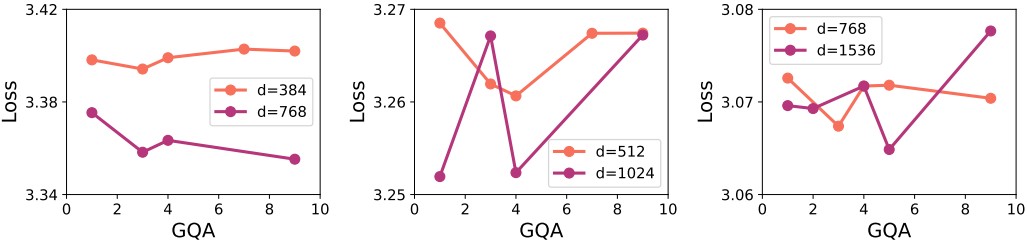

Figure 24: **Loss vs. GQA.** (left) 80M model variants; (center) 145M model variants; (right) 297M model variants. Across different model sizes, the relationship between training loss and GQA varies substantially when hidden size and the mlp-to-attention ratio are fixed. The legend denotes the hidden size of each trained model.

## J MORE ABLATION STUDY

In this section, We first evaluate the impact of outlier data on the fitting of the scaling laws in Figure 25 (left) and Figure 25 (center). Then, we evaluate the fitting performance of multiplicative calibrations and additive calibrations in Figure 25 (left) and Figure 25 (right).

Finally, we evaluate the performance of Joint and non-separable calibrations shown below in Figure 26:

$$(a_0 + a_1 \log(\frac{dr}{\sqrt{N}}) + a_2/(\frac{dr}{\sqrt{N}})) \cdot L_{\text{opt}}$$

where $d = d_{\text{model}}$, $r = r_{\text{mlp/attn}}$, and $N = N_{\text{non-embed}}$. In Figure 26, we observe that the performance of joint and non-separable calibrations is significantly worse than that of multiplicative calibration, consistent with our discussion in §3.3.

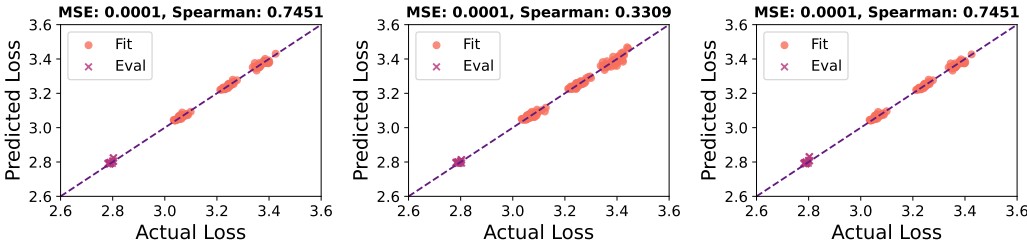

Figure 25: **Ablation Study.** (left) use multiplicative calibrations without outliers; (center) use multiplicative calibrations with outliers; (right) use additive calibrations without outliers. The outlier refers to models trained with an mlp-to-attention ratio below 0.5 or above 5. We observe that outlier data points harm the scaling law fit. Moreover, while multiplicative and additive calibrations differ in formulation, their MSE and Spearman values remain nearly identical. Dots denote the data points used for fitting, while crosses indicate the test data points.

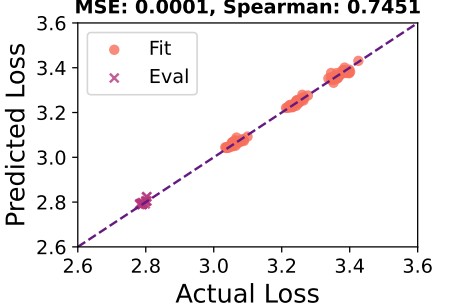
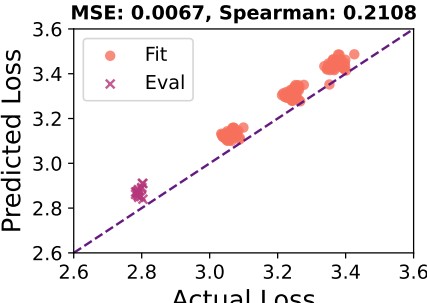

Figure 26: **Joint and non-separable calibrations.** (left) use multiplicative calibrations; (right) use joint and non-separable calibrations. We observe that joint and non-separable calibrations yield higher MSE and lower Spearman scores than multiplicative calibrations, indicating inferior performance. Dots denote the data points used for fitting, while crosses indicate the test data points.

# K INFERENCE FLOPS ANALYSIS

Building on the inference FLOPs analysis from prior work Kaplan et al. (2020), we begin with the following definition:

- $d_{\text{model}}$: hidden size
- $f_{\text{size}}$: intermediate (feed-forward) size
- $n_{\text{layers}}$: number of layers
- $A$: number of query heads
- $K$: number of key/value heads
- $d_h$: per-head hidden dimension (query and value)
- $T$: per-head hidden dim the KV length prior to token generation

Based on the above definition, we have $d_q = Ad_h$ and $d_{kv} = Kd_h$. We focus exclusively on non-embedding FLOPs, resulting in:

Attention: QKV and Project

$$n_{\text{layers}}(\underbrace{2d_{\text{model}}d_q}_{Q} + \underbrace{2d_{\text{model}}d_{kv}}_{K} + \underbrace{2d_{\text{model}}d_{kv}}_{V} + \underbrace{2d_{\text{model}}d_q}_{O})$$

Attention: Mask

$$n_{\text{layers}}(2Td_q)$$

Feedforward:

$$n_{\text{layers}}(3 \cdot 2d_{\text{model}}f_{\text{size}})$$

Total Inference non-embedding FLOPs:

$$\text{Total-FLOPs} = n_{\text{layers}}(\underbrace{2d_{\text{model}}d_q}_{Q} + \underbrace{2d_{\text{model}}d_{kv}}_{K} + \underbrace{2d_{\text{model}}d_{kv}}_{V} + \underbrace{2d_{\text{model}}d_q}_{O} + \underbrace{2Td_q}_{qK^{\top}} + \underbrace{3 \cdot 2d_{\text{model}}f_{\text{size}}}_{\text{up, gate, down}})$$

Since $P_{\text{non-emb}} \approx n_{\text{layers}}(2d_{\text{model}}d_q + 2d_{\text{model}}d_{kv} + 3d_{\text{model}}f_{\text{size}})$. Therefore, Total-FLOPs $= 2P_{\text{non-emb}} + 2n_{\text{layers}}Td_q$

we adopt the following three approaches to accelerate inference:

- Increasing the MLP-to-Attention ratio reduces the term $2Td_q$, thereby lowering the total FLOPs.
- Increasing the hidden size reduces the term $2Td_q$, thereby lowering the total FLOPs.

## L   MORE LARGE-SCALE TRAINING RESULTS

In this section, we first show the detailed result over downstream tasks of large-scale models in Table 9 and Table 10.

Table 9: **Detailed Results on Downstream Tasks for 1B Models.** In this table, we show detailed results of 1B models over 9 downstream tasks.

| Downstream Tasks | LLaMA-3.2-1B | Panda-1B | Surefire-1B |
|---|---|---|---|
| Arc-Easy | 58.8 | 60.9 | 59.7 |
| Arc-Challenge | 29.8 | 28.9 | 30.2 |
| LAMBADA | 52.8 | 55.1 | 52.0 |
| HellaSwag | 56.9 | 58.4 | 56.6 |
| OpenBookQA | 32.0 | 33.2 | 32.0 |
| PIQA | 73.6 | 75.2 | 73.0 |
| SciQ | 84.8 | 87.2 | 84.9 |
| WinoGrande | 57.1 | 58.6 | 57.5 |
| COQA | 48.7 | 55.3 | 52.7 |
| Avg. | 54.9 | 57.0 | 55.4 |

Table 10: **Detailed Results on Downstream Tasks for 3B Models.** In this table, we show detailed results of 3B models over 9 downstream tasks.

| Downstream Tasks | LLaMA-3.2-3B | Panda-3B | Surefire-3B | Panda-3B$^\circ$ |
|---|---|---|---|---|
| Arc-Easy | 66.4 | 65.5 | 67.6 | 66.8 |
| Arc-Challenge | 33.3 | 35.2 | 33.9 | 33.3 |
| LAMBADA | 60.6 | 61.8 | 61.4 | 61.5 |
| HellaSwag | 66.7 | 66.9 | 67.0 | 67.8 |
| OpenBookQA | 38.4 | 38.6 | 38.6 | 38.0 |
| PIQA | 76.8 | 76.9 | 77.4 | 76.8 |
| SciQ | 89.4 | 91.2 | 92.1 | 90.5 |
| WinoGrande | 62.5 | 63.2 | 60.5 | 62.7 |
| COQA | 63.3 | 63.4 | 65.4 | 64.9 |
| Avg. | 61.9 | 62.5 | 62.6 | 62.5 |

# M MoE Inference

In this section, we examine how the Mixture-of-Experts (MoE) architecture affects inference efficiency. Figure 27 indicates that larger hidden sizes and higher Active-Experts-to-Attention ratios improve the inference throughput of MoE models, consistent with observations in dense models.

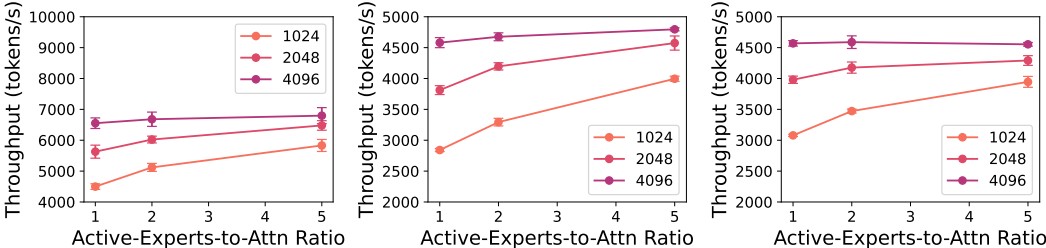

Figure 27: **Active-Experts-to-Attn on Inference Throughput.** (left) 3B-A1.1B model variants; (center) 5.3B-A1.7B model variants; (right) 8.3B-A1.5B model variants. We study the effect of the Active-Experts-to-Attention ratio on inference throughput by fixing the total number of active parameters, setting GQA to 4, and using a batch size of 2048 to reduce MoE inference variance in this figure. All evaluations are performed using the vLLM framework Kwon et al. (2023) on a single NVIDIA Ampere 40GB A100 GPU with 1024 input and 256 output tokens.

