# OpenReview forum: "Scaling Laws Meet Model Architecture: Toward Inference-Efficient LLMs"
_ICLR.cc/2026/Conference — ICLR 2026 Poster_

### Official Review · Reviewer_qzeo · 2025-10-20

**Soundness:** 3
**Presentation:** 3
**Contribution:** 2
**Rating:** 6
**Confidence:** 3

**Summary:**

This paper addresses a challenge in the efficient deployment of large language models. Whiile existing scaling laws guidfe the trade-off between model size and training data for optimal accuracy, they largely ignore inference efficiency. This work bridges the gap by investigating how architecture hyperparameters impact inference throughput and model accuracy, including hidden size, mlp-to-attention ratio, and GQA.

**Strengths:**

1. **Novel and practical contribution**: The focus on inference scaling law is highly relevant for the practical deployment of LLMs. The proposed conditional scaling law is a clever and effective extension to the scaling law.
2. **Rigorous and extensive experiments**: The scale of the study is impressive, with over 200 models trained across different parameter scales. This provides a solid empirical foundation for their claims.

**Weaknesses:**

1. **The design for the conditional scaling law is lack of theoretical support**. The paper doesn't present why the conditional scaling law is fomulated as a multiplicative or additive equation. Why these two forms act similar? Are they still similar if the candidate models are much larger (e.g., 70B+)? Which one should we take in the future research?
2. **Lack of a clear summarization**: It is encouraged to present an summarization on the conclusions about how to design an efficient LLM architecture. It is also encouraged to present an comparison on the design choices of existing open source models about which models are closer to the optimal design.
3. **Lack of analysis on GQA**: Why the relationship between loss and GQA is highly fluctuating? This conclusion is counterintuitive and requires a discussion.

**Questions:**

The questions are listed in the Weaknesses. I believe this direction is worth researching and this paper will be a good start if the above problems are well solved.

---

> ### Author Response · Authors · 2025-12-02
>
> Thank you for your careful evaluation of our work and your helpful, constructive feedback.
>
> **Q1: Design of conditional scaling law lacks theoretical support**
>
> **Response:** This is an important question and reflects a broader limitation shared by many empirical scaling-law studies, including Chinchilla (Hoffmann et al., 2022, Eq. 1) and more recent formulations such as Tian et al. (2025, Eq. 4) and Chen et al. (2025, Eq. 4), where multiple functional forms are proposed without strong theoretical grounding. In our work, we focus on two simple and transparent calibration forms, multiplicative and additive, and our ablations of multiplicative, additive, and even more complex joint, non-separable formulations (Lines 367-373) show simple calibrations achieve very similar MSE and Spearman correlations. The two-step reference-and-calibration framework appears robust enough that simple calibrations perform well, with potential differences arising in the presence of data or model outliers. We recommend using the multiplicative form as a practical default.
> Regarding much larger models (e.g., 70B+), additional experiments would certainly be valuable. Nevertheless, we expect the approach to remain applicable for model families with consistent architectural patterns such as LLaMA-3.2 and Qwen. We anticipate that applying the method to MoE models will be more challenging due to their additional architectural factors and will require more extensive experimentation.
>
> Hoffmann et al., 2022. Training Compute-Optimal Large Language Models
>
> Tian et al., 2025. Towards Greater Leverage: Scaling Laws for Efficient Mixture-of-Experts Language Models
>
> Chen et al, 2025. Scaling Law for Quantization-Aware Training
>
> **Q2: Lack of a clear summarization**
>
> **Response:** Thank you for the suggestion. We summarize our procedure for identifying inference-efficient accurate models in Algorithm 1 using a two-step conditional framework: (i) train small models to fit the conditional scaling law (Eq. 3), and (ii) solve Eq. 4 for the predicted optimal architecture followed by a local search over GQA to maximize inference efficiency. We also provide a clearer high-level summary in the Conclusion section.
>
> We additionally compare design choices across existing open-source models at the 1B and 3B scales. Alongside the LLaMA-3.2-1B, Panda-1B, and Surefire-1B models we pretrained, we report inference throughput (tokens per second), byte-level WikiText PPL, and full architectural configurations in the accompanying tables. All throughput measurements are performed with vLLM on H200 GPUs using batch size 128. For the **1B** scale, we include LLaMA-3.2-1B-HF and OLMo-2-0425-1B-HF. Because OLMo supports only a 4k context window and cannot run our standard 4k-in/1k-out setup, we also report results for a 2k-in/1k-out setting. For the **3B** scale, we add LLaMA-3.2-3B-HF and Qwen2.5-3B-HF, all evaluated under the 4k-in/1k-out configuration. We present these results in Tables 7 and 8 of Appendix G.
>
> Our observations are as follows:
>
>  (i) OLMo-2-0425-1B-HF is relatively close to our predicted optimal design, with an MLP-to-attention ratio of 3 (near our predicted 3.6), but remains inference-inefficient due to its hidden dimension and GQA choices.
>
>  (ii) At the 3B scale, LLaMA-3.2-3B-HF achieves good accuracy but is not inference-efficient, while Qwen2.5-3B-HF is inference-efficient but less accurate. These comparisons highlight the need for, and relevance of, our inference-efficient accurate model designs.
>
> **1B Models:**
>
> |  | LLaMA-3.2-1B | Panda-1B | Surefire-1B | LLaMA-3.2-1B-HF | OLMo-2-0425-1B-HF |
> |:---:|:---:|:---:|:---:|:---:|:---:|
> | Wikitext PPL | 1.7151 | 1.7016 | 1.7142 | 1.5807 | 1.5798 |
> | Tput (4k/1k) | 9306 | 6218 | 11283 | 9306 | / |
> | Tput (2k/1k) | 11948 | 8961 | 13890 | 11948 | 7486 |
> | **Model Architectural Config** | | | | | |
> | num layers | 16 | 16 | 16 | 16 | 16 |
> | Hidden size | 2048 | 2560 | 2560 | 2048 | 2048 |
> | MLP-to-Attn ratio | 4.8 | 1.067 | 3.6 | 4.8 | 3 |
> | GQA  | 4 | 4 | 9 | 4 | 1 |
> | Non-emb Params  | 973M | 975M | 965M | 973M | 1.074B |
>
> **3B Models**
>
> |  | LLaMA-3.2-3B | Panda-3B | Surefire-3B | LLaMA-3.2-3B-HF | Qwen2.5-3B-HF |
> |:---:|:---:|:---:|:---:|:---:|:---:|
> | Wikitext PPL | 1.6489 | 1.6454 | 1.6462 | 1.5164 | 1.6185 |
> | Tput (4k/1k) | 4311 | 3335 | 4842 | 4311 | 6470 |
> | **Model Architectural Config** | | | | | |
> | num layers | 28 | 28 | 28 | 28 | 36 |
> | Hidden size | 3072 | 4096 | 4096 | 3072 | 2048 |
> | MLP-to-Attn ratio | 3 | 1 | 1 | 3 | 7.167 |
> | GQA  | 3 | 3 | 7 | 3 | 8 |
> | Non-emb Params  | 2.82B | 2.82B | 2.82B | 2.82B | 2.77B |

---

> > ### Author Response · Authors · 2025-12-02
> >
> > **Q3: Lack of analysis on GQA**
> >
> > **Response:** Please see our general response **GQ3**. In brief, with **the total number of heads fixed** (i.e., = Q-heads + KV-heads), increasing the number of Q-heads expands the query projection but simultaneously reduces the number of KV-heads, increasing KV sharing and thus reducing KV expressivity. Conversely, decreasing Q-heads increases KV-head capacity (fewer replicas) but reduces the projection dimensionality of both Q and KV. These opposing effects create a tradeoff, making the relationship between GQA and training loss non-smooth and often highly fluctuating.

---

### Official Review · Reviewer_Bn1c · 2025-10-27

**Soundness:** 4
**Presentation:** 3
**Contribution:** 3
**Rating:** 8
**Confidence:** 4

**Summary:**

This paper revisits scaling laws for large language models (LLMs) through the lens of model architecture and inference efficiency.
While existing works (e.g., Chinchilla) focus on training efficiency by relating model parameters, data tokens, and loss, this work introduces architecture-aware scaling laws that explicitly incorporate structural hyperparameters affecting inference cost.

Through a large empirical study (>200 models from 80M–3B parameters trained on 8–100B tokens), the paper shows that performance and inference throughput follow predictable trends governed by these architecture variables. Using these fits, the authors predict “optimal” configurations for new small models (e.g., Panda-1B, Panda-3B) that achieve ~26% higher inference throughput and ~2% higher accuracy than LLaMA-3.2-1B.

**Strengths:**

1. Novel perspective: Introduces an architecture-aware formulation of scaling laws, bridging the gap between training efficiency and inference efficiency.
2. Extensive empirical coverage: The study includes over 200 model configurations across hidden sizes, attention/MLP ratios, and token budgets.
3. Practical insight: Demonstrates that scaling law fits from small models generalize to larger ones, potentially guiding efficient architecture search.

**Weaknesses:**

1. Empirical “optimality” is descriptive: The proposed “conditional scaling law” is fitted post-hoc to data rather than derived from optimization principles.
2. Hardware dependence: Reported inference efficiency improvements rely on specific implementation details (vLLM, A100), which may not generalize to other hardware or serving stacks.

**Questions:**

1. How sensitive are the predicted “optimal” ratios to the hardware platform (A100 vs. H100 or TPU)? Could authors provide several experiments on another hardware?

---

> ### Author Response · Authors · 2025-12-02
>
> We appreciate your thoughtful review and the constructive suggestions you provided.
>
> **Q1: Empirical “optimality” is fitted post-hoc to data rather than derived from optimization principles**
>
> **Response:** We formalize our approach as an explicit optimization problem in Lines 267-268 (Eq. 4), following the similar formulation used in the original Chinchilla scaling study (Hoffmann et al., 2022, Eq. 1) and more recent scaling-law work such as Tian et al., 2025 (Eq. 4) and Chen et al., 2025 (Eq. 4). Regarding the concern that the “optimality” is fitted post hoc rather than derived from first principles, we note that this is precisely the purpose and standard methodology of empirical scaling laws: to estimate relationships from small-scale pretraining experiments and use these empirical fits to guide predictions and extrapolations to larger models. Our method aligns with the established practice introduced in the Chinchilla framework and consistently followed by subsequent scaling-law literature.
>
> Hoffmann et al., 2022. Training Compute-Optimal Large Language Models
>
> Tian et al., 2025. Towards Greater Leverage: Scaling Laws for Efficient Mixture-of-Experts Language Models
>
> Chen et al, 2025. Scaling Law for Quantization-Aware Training
>
> **Q2: Inference efficiency across stacks and hardware**
>
> **Response:** Please see our general response GQ2. As presented in Appendix E&F, the new results with H200 or SGLang are consistent with our original (vLLM, A100) evaluations. In Figure 6, Surefire-1B and Surefire-3B outperform LLaMA-3.2-1B and LLaMA-3.2-3B across all settings, achieving up to 47% higher inference throughput with SGLang on H200s, compared to 42% with vLLM on A100s.
>
> **Q2.1: sensitivity of the predicted “optimal” to hardware**
>
> **Response:** We would like to note that inference efficiency depends on the specific hardware and serving stack, whereas pretraining outcomes such as the training loss should not. Under a fixed dataset, training procedure, numeric precision, and source of randomness, the training loss is expected to remain consistent across hardware platforms. If different hardware configurations yield meaningfully different pretraining outcomes, this usually points to implementation issues or numerical instabilities rather than inherent hardware effects. In practice, we treat results from NVIDIA GPUs as our primary reference due to their widespread use, mature software ecosystem, and well-validated numerical behavior.

---

### Official Review · Reviewer_rBcz · 2025-10-30

**Soundness:** 2
**Presentation:** 3
**Contribution:** 2
**Rating:** 4
**Confidence:** 4

**Summary:**

This paper addresses the critical and underexplored trade-off between the accuracy and inference efficiency of Large Language Models (LLMs). While traditional scaling laws focus on training compute, parameters (N), and data (D), they largely ignore the architectural choices that significantly impact inference cost, which is a dominant expense in real-world deployment.

**Strengths:**

The proposed conditional scaling law is a novel extension of established Chinchilla scaling laws12. By formulating the loss as a separable, calibrated function of architectural ratios relative to a Chinchilla-optimal reference loss, the authors provide a practical tool for architecture design.

Besides, the claims are substantiated by a large-scale empirical study involving over 200 trained models, spanning a wide range of parameter counts (80M to 3B) and token budgets (8B to 100B). This comprehensive dataset provides a strong foundation for fitting the scaling laws.

**Weaknesses:**

1. The experiments are validated up to 3B parameters. While this is a substantial undertaking, the authors acknowledge that the evaluation does not extend to the 7B+ scale, which is a very common size for deployed open-source models.
2. The entire analysis is confined to dense decoder-only transformers. It is unclear if these findings, particularly the U-shaped loss curves and the separability assumption, would hold for other popular architectures like Mixture-of-Experts (MoE).
3. The study finds that GQA significantly impacts inference efficiency (Figure 9) but has a "highly fluctuating" and inconsistent relationship with training loss (Figure 13). Consequently, GQA is not integrated into the conditional scaling law itself. Instead, it is handled via a separate "local search" after other parameters are optimized. This feels less principled and integrated.

**Questions:**

The authors note the "highly fluctuating" relationship between GQA and loss (Figure 13)33. Do you have a hypothesis for why GQA behaves so erratically compared to the smooth curves for $d_{model}$ and $r_{mlp/attn}$?

---

> ### Author Response · Authors · 2025-12-02
>
> Thanks for taking the time to review our paper and providing constructive feedback.
>
> **Q1: Scaling experiments beyond 3B**
>
> **Response:** Please see our general response **GQ1**, where we augmented our study with training an inference-efficient 3B model, Surefire-3B, with results in Tables 1 and 2. The two-step conditional method to predict the optimal architecture can be naturally extended to the 7B scale via fitting the law over 1B/3B trained models. Since training a single 7B+ model to 200B would require approximately **28 days** on our limited compute resources (64 A100 GPUs), therefore, it cannot be completed within the discussion period.
>
> **Q2: Hypothesis for "highly fluctuating" GQA**
>
> **Response:** Please see our general response **GQ3**. In brief, with **the total number of heads fixed** (i.e., = Q-heads + KV-heads), increasing the number of Q-heads expands the query projection dimensionality but simultaneously reduces the number of KV-heads, increasing KV sharing and thus reducing KV expressivity. Conversely, decreasing Q-heads increases KV-head capacity (fewer replicas) but reduces the projection dimensionality of both Q and KV. These opposing effects create a tradeoff, making the relationship between GQA and training loss non-smooth and often highly fluctuating.
>
> **Q3: Extension to other popular architectures like MoE**
>
> **Response:** We are also very interested in applying our method to MoE models. However, MoE architectures introduce substantially more interacting design factors than dense models, such as the number of active experts (topK), the total number of experts, and the choice of routing mechanism. Meaningful conclusions often require training larger MoE models, which makes comprehensive and controlled evaluation more challenging. We therefore leave a full, stable, and systematic exploration of MoE architectures to future work.

---

### Official Review · Reviewer_fQwj · 2025-11-03

**Soundness:** 3
**Presentation:** 3
**Contribution:** 3
**Rating:** 8
**Confidence:** 3

**Summary:**

This work builds on the Chinchilla scaling laws by adding architecture-based dimensions like hidden size, MLP-to-attention ratios, and grouped-query attention, to analyse the trade-off between inference efficiency and downstream accuracy. Authors fit a conditional scaling law on >200 Llama-style models (80M-30B #params), authors identify Pareto-optimal configurations and improve Panda/Surefire models that improve downstream accuracy and inference throughput over Llama-3.2 baselines.

**Strengths:**

- Conditioning the Chinchilla scaling laws on hidden size and MLP-to-attention ratios is timely, useful, and interesting
- Experiments over 200+ trained models provide robust empirical results
- Architectures suggested by the new scaling laws yield sensible gains -- Panda models raise average zero-short accuracy, while Surefire significantly improves the inference throughput by up to ~25%

**Weaknesses:**

- Results stop at 3B parameters -- what happens at larger model scales?

**Questions:**

- What happens for larger models? Do the authors have architectural recommendations for these regimes?
- The paper's results are based on vllm -- do results transfer also to e.g. SGLang (https://github.com/sgl-project/sglang)?

---

> ### Author Response · Authors · 2025-12-02
>
> We appreciate your time in reviewing our paper and your constructive comments.
>
> **Q1: Scaling experiments beyond 3B**
>
> **Response:** Please see our general response **GQ1**, where we augmented our study with training an inference-efficient 3B model, Surefire-3B, with results in Tables 1 and 2. The two-step conditional method to predict the optimal architecture can be naturally extended to the 7B scale via fitting the law over 1B or 3B trained models. Since training a single 7B+ model to 200B would require approximately **28 days** on our limited compute resources (64 A100 GPUs), therefore, it cannot be completed within the discussion period.
>
> **Q2: Inference efficiency on SGLang**
>
> **Response:** Please see our general response **GQ2**. As presented in Appendix E&F, the new results with SGLang are consistent with our original vLLM evaluations. In Figure 14, Surefire-1B and Surefire-3B outperform LLaMA-3.2-1B and LLaMA-3.2-3B across all settings, achieving up to **30%** higher inference throughput with SGLang on A100s, compared to **42%** with vLLM on A100s. Results on H200 are also provided in Figure 18.

---

### Author Response · Authors · 2025-12-02
**General Response**

We would like to thank all reviewers for their constructive comments and helpful feedback. Reviewers commend our work as a **novel, timely, and architecture-aware** extension of Chinchilla scaling laws, supported by 200+ trained models. They also highlight the practical value of the conditional formulation for designing **inference-efficient** architectures, with derived models achieving clear accuracy gains and up to ~47% throughput improvement over the 3B models when evaluated on H200 GPUs using the SGLang framework. Here we address some shared questions and weaknesses from reviewers:

**GQ1: Scaling experiments beyond 3B (reviewers fQwj, rBcz)**

**Response:** We have augmented our study with an inference-efficient 3B model, Surefire-3B, which matches the accuracy of LLaMA-3.2-3B while achieving up to **42%** higher inference throughput (vLLM on A100). The new results are now included in Tables 1 and 2. While we agree that 7B+ evaluations would further strengthen the study, training a single 7B+ model to 200B would require approximately **28 days** on our limited compute resources (64 A100 GPUs), and therefore cannot be completed within the discussion period.

**GQ2: Inference efficiency across stacks and hardware (reviewers fQwj, Bn1c)**

**Response:** We have expanded our inference efficiency analysis to include both vLLM and SGLang across A100 and H200 hardware, as presented in Appendix E&F. The results are consistent with our original (vLLM, A100) evaluations, indicating that the observed efficiency gains transfer across serving stacks and hardware configurations. As shown in Table 6 of Appendix G, Surefire-1B and Surefire-3B outperform LLaMA-3.2-1B and LLaMA-3.2-3B across all settings, achieving up to **47%** higher inference throughput with SGLang on H200s, compared to **42%** with vLLM on A100s.

**GQ3: Analysis of GQA’s fluctuating relationship with training loss (reviewer rBcz, qzeo)**

**Response:** Varying GQA does not adjust model capacity in the coordinated way that changing $d_{\text{model}}$ or $r_{\text{mlp/attn}}$ does, where the dimensions of (Q), (K), and (V) scale together predictably. In order to keep the number of attention parameters fixed, we vary GQA by **holding the total number of heads fixed** (i.e., total heads = Q-heads + KV-heads) and **reallocating** this fixed budget between query and key-value heads, producing asymmetric changes in their effective dimensionalities. Prior work shows only that Q/K/V projections can have non-interchangeable roles (e.g., head-importance heterogeneity, Voita et al., 2019), but provides no monotonic or predictive theory for how reallocating capacity across Q versus KV should affect loss. Consistent with this, recent open LLMs choose **different GQA settings even within a single family**: Qwen3 uses GQA = 2 for 0.6B/1.7B, GQA = 4 for 4B/8B, GQA = 5 for 14B, GQA = 8 for 32B, and for the 30B-A3B MoE; LLaMA-3/3.1/3.2 likewise use GQA = 4, 8, and 3 across closely related sizes. This variation across models of similar architecture shows that GQA is treated as a **discrete, model-specific hyperparameter**, supporting our decision to tune it via local search rather than integrate it into the continuous scaling law. We added this analysis to Appendix H.

Voita et al., 2019, Analyzing Multi-Head Self-Attention: Specialized Heads Do the Heavy Lifting.

We have updated the PDF with the following revisions in response to the reviewers’ requests:
* Pretrained and added an inference-efficient 3B model, Surefire-3B (Table 1).
* Added an ablation of the fitting-data strategy in Section 5 (line 466), to make the fitting of the conditional scaling law more efficient, pretrained Panda-3B$^{\circ}$ using only 1B data, achieving comparable performance (Table 2, Figure 7).
* Included analysis of GQA’s fluctuating relationship with training loss in Appendix H.
* Added an ablation of inference efficiency across hardware and serving stacks, covering (vLLM, SGLang) × (A100, H200) in Appendix G.
* Expanded the calibration-scheme ablation in Section 5.
* Included a high-level summary of our method in the conclusion and a comparison with existing open models in the appendix.

---

### Meta-Review · Area_Chair_bxgc · 2025-12-31

**Summary:**

The paper receives scores of 8, 8, 6, and 4, so the overall feedback is more positive than negative. Several strengths are mentioned, including that the proposed conditional scaling law is novel, useful, and interesting. The main concerns from the 4 (Reviewer rBcz) are: (1) experiments are only validated up to 3B parameters, which is not large enough (Reviewer fQwj raised the same concern); (2) the analysis is confined to dense decoder-only transformers and does not cover MoE models; and (3) GQA shows a "highly fluctuating" and inconsistent relationship with training loss (Reviewer qzeo raised a similar question). Evaluating larger models would indeed strengthen the validation of the paper's findings and conclusions. Other common concerns include potential dependence on specific hardware and serving stacks, since the results are based on vLLM, it is unclear whether they transfer to other serving frameworks such as SGLang, as well as a lack of theoretical support.

**Reviewer Concerns:**

For Reviewers rBcz and fQwj, the key issue is that experiments only go up to 3B parameters. The authors explained that, due to current computing constraints, they could not run larger-scale experiments, and it is encouraged to include these results in a future version. I believe most of the other concerns have likely been addressed in the rebuttal.

**Reviewer Scores:**

It is difficult to say whether reviewers will update their scores, Reviewer rBcz is likely to revise the score upward.

---

### Decision · Program_Chairs · 2026-01-26

Accept (Poster)